# Is Learning in Games Good for the Learners?

**William Brown**
Columbia University
w.brown@columbia.edu

**Jon Schneider**
Google Research
jschnei@google.com

**Kiran Vodrahalli**
Google Research
kirannv@google.com

## Abstract

We consider a number of questions related to tradeoffs between reward and regret in repeated gameplay between two agents. To facilitate this, we introduce a notion of *generalized equilibrium* which allows for asymmetric regret constraints, and yields polytopes of feasible values for each agent and pair of regret constraints, where we show that any such equilibrium is reachable by a pair of algorithms which maintain their regret guarantees against arbitrary opponents. As a central example, we highlight the case one agent is no-swap and the other's regret is unconstrained. We show that this captures an extension of *Stackelberg* equilibria with a matching optimal value, and that there exists a wide class of games where a player can significantly increase their utility by deviating from a no-swap-regret algorithm against a no-swap learner (in fact, almost any game without pure Nash equilibria is of this form). Additionally, we make use of generalized equilibria to consider tradeoffs in terms of the opponent's algorithm choice. We give a tight characterization for the maximal reward obtainable against *some* no-regret learner, yet we also show a class of games in which this is bounded away from the value obtainable against the class of common "mean-based" no-regret algorithms. Finally, we consider the question of learning reward-optimal strategies via repeated play with a no-regret agent when the game is initially unknown. Again we show tradeoffs depending on the opponent's learning algorithm: the Stackelberg strategy is learnable in exponential time with any no-regret agent (and in polynomial time with any no-*adaptive*-regret agent) for any game where it is learnable via queries, and there are games where it is learnable in polynomial time against any no-swap-regret agent but requires exponential time against a mean-based no-regret agent.

## 1 Introduction

How should two rational agents play a repeated, possibly unknown, game against one another? One natural answer – barring any knowledge of the game, or the capacity to compute potentially computationally intractible equilibria – is that they should employ some sort of learning algorithm to learn how to play over time. Indeed, there is a vast literature which studies what happens when all the players in a repeated game run (some specific type of) learning algorithms to select their actions. For example, when all players in a game simultaneously run no-(swap)-regret learning algorithms, it is known that the average strategy profile of the learners converges to a (coarse) correlated equilibrium ([23]; [11]; [16]). More recent works have studied how to design algorithms that converge to these equilibria at faster rates ([7]; [1]; [8]), performance guarantees of such equilibria compared to the optimal possible welfare ([4]; [17]), and the specific dynamics of such learning algorithms ([9]; [20]; [22]).

In contrast, relatively little attention has been devoted to whether it is actually *in the interest of these agents* to run these specific learning algorithms. For example, in a setting where all agents are running no-swap-regret learning algorithms, when can an agent significantly benefit by deviating and running a different type of algorithm? And if they can, what algorithm should the agent deviate to?

37th Conference on Neural Information Processing Systems (NeurIPS 2023).

## 1.1 Our results

We explore the following questions (and others) in the case where two agents repeatedly play a normal-form, general-sum game for $T$ rounds.

**When does reward trade off with regret?** While maximizing reward is often viewed as the objective of regret minimization against arbitrary adversaries, tensions may emerge when playing against another learning agent, as one's choice of actions has a causal effect on future loss functions as determined by the opponent's algorithm. In such cases, as we discuss further below, it turns out that demanding a stronger regret guarantee (e.g. asking for no-swap-regret instead of no-external-regret) may ultimately result in lower reward for an agent. To analyze these tradeoffs, in Section 2 we introduce a notion of *generalized* $(\Phi_A, \Phi_B)$-*equilibrium*, where $\Phi_A$ and $\Phi_B$ are the sets of "deviation strategies" over which players $A$ and $B$ minimize regret (e.g. fixed actions or swap functions), presented as an asymmetric extension of the linear $\Phi$-equilibria considered by (13). Each pair of strategy sets $(\Phi_A, \Phi_B)$ generates a polytope of $(\Phi_A, \Phi_B)$-equilibria, where any point then yields a reward value for each player. We show that all such points are feasible: for any game and any $(\Phi_A, \Phi_B)$-equilibrium $\varphi$, there is a pair of algorithms which converge to $\varphi$ while maintaining their respective $\Phi_A$-regret and $\Phi_B$-regret guarantees against arbitrary opponents. Further, deviating to a strategy with fewer constraints $\Phi_A$ can often result in *strictly* improved reward for player $A$. As a concrete application, we consider the question of deviating from simultaneous no-swap-regret play.

**When is no-swap-regret play a stable equilibrium?** What should you do if you know that your opponent in a repeated game is running a no-swap-regret algorithm to select their actions? In (10), the authors show that one utility-optimizing response (up to additive $o(T)$ factors) is to play a static (mixed) strategy (your *Stackelberg strategy*) and obtain the *Stackelberg value* of the game. However, determining your Stackelberg strategy requires some knowledge of the game, and acquiring this knowledge from repeated play may be difficult (we address the latter issue in Section 5). In comparison, it is relatively straightforward to also run a no-swap-regret learning algorithm. This begs the question: are there games where you obtain significantly less utility (i.e., at least $\Omega(T)$ less utility) by running a no-swap-regret learning algorithm instead of playing your Stackelberg strategy?

We show that the answer is *yes*, such games exist and are relatively common. In fact, we provide an efficient algorithmic characterization of the games $G$ for which both players playing a no-swap-regret learning algorithm is an ($o(T)$-)approximate Nash equilibrium of the entire repeated "meta-game". The exact characterization is presented in Section 3 and is somewhat subtle, e.g., there are slightly different characterizations depending on whether we insist all pairs of no-swap-regret algorithms lead to approximate equilibria or only one specific pair, corresponding to best-case and worst-case values in the generalized equilibrium polytope. One consequence of both characterizations, however, is that for *almost all* games (in a measure-theoretic sense, considering arbitrarily small perturbations), in order for it to be an approximate equilibrium for both players to play a low-swap regret strategy, the game $G$ must possess a *pure* Nash equilibrium. That is, in any game without a pure Nash equilibrium, it is possible for at least one of the parties to do significantly better by switching from no-swap-regret learning to playing their Stackelberg strategy.

Finally, we additionally show there are some games where playing a no-(external)-regret algorithm against another no-swap-regret learner weakly dominates playing a no-swap-regret algorithm, regardless of the specific choice of algorithms. This counters the intuition that stronger regret guarantees protect a player from worse outcomes.

**Optimizing reward against no-regret learners.** What if our opponent is not running a no-swap-regret algorithm, but simply a no-(external)-regret algorithm? In this case, it is still possible to obtain at least the Stackelberg value of the game by playing our Stackelberg strategy (no-regret algorithms are also guaranteed to eventually learn and play the best response to this strategy). However, unlike in the no-swap-regret setting, there exist specific games and no-regret algorithms where it is possible to obtain *significantly* ($\Omega(T)$) more than the Stackelberg value by playing a specific dynamic strategy. This phenomenon was first observed in (10), where a specific game is given for which it is possible to obtain $\Omega(T)$ more than Stackelberg when playing against any no-regret algorithm in the family of *mean-based learning algorithms* (including algorithms such as multiplicative weights and EXP3). However, many questions remain unanswered, such as understanding in which games it is possible to outperform playing one's Stackelberg strategy, and by how much.

In Section 4 we present some answers to these questions for the case of generic no-regret algorithms. Specifically, we first show that if player $B$ is running (any) no-regret algorithm, the utility of player $A$ (regardless of what strategy they employ) is upper bounded by $\mathsf{Val}_A(\emptyset, \mathcal{E}) \cdot T + o(T)$, where $\mathsf{Val}_A(\emptyset, \mathcal{E})$ is what we call the the *unconstrained-external (equilibrium) value* of the game for player $A$, which is given by the solution to a linear program. We then show that this upper bound is asymptotically tight: there exists a no-regret algorithm $\mathcal{L}$ such that if player $B$ is playing according to $\mathcal{L}$, then the player $A$ can obtain utility at least $\mathsf{Val}_A(\emptyset, \mathcal{E}) \cdot T - o(T)$ by playing an appropriate strategy in response.

Note that this characterization does not completely resolve the question of (10) – it requires the construction of a fairly specific no-regret algorithm $\mathcal{L}$, and it is still open what is possible against specific (classes of) no-regret algorithms (e.g., multiplicative weights or mean-based algorithms). In fact, we show a property of games which, when satisfied, implies that it is impossible to obtain the unconstrained-external value against a mean-based learner.

**Learning the Stackelberg strategy through repeated play.** Finally, we address the question of how hard it is to actually identify the Stackelberg strategy in a game against a learning opponent. Given full knowledge of the game $G$, finding an agent's Stackelberg strategy is simply a computational problem which turns out to be efficiently solvable by solving several small LPs (see (6)). However, if the game (in particular, the opponent's reward matrix) is unknown, an agent must learn the Stackelberg strategy over time. Existing work on learning Stackelberg strategies (e.g., (19; 24)) generally assumes access to a *best-response oracle* for the game (i.e., for a specific mixed strategy, how will an opponent best-respond?). In contrast, if our opponent is playing a specific no-regret learning algorithm, they may not immediately best respond to the strategies we play! This raises the following two questions. First, when is it possible to learn the Stackelberg equilibrium of a game while playing against a learning opponent? Second, is it easier to learn this equilibrium when playing against certain classes of learning algorithms?

In Section 5 we begin by showing that it is indeed possible to convert any best-response query algorithm for finding Stackelberg equilibria via best-response queries to an adaptive strategy that learns Stackelberg equilibria via repeated play against a generic no-regret learner, albeit potentially at the cost of an exponential blow-up in the number of rounds, e.g. for a query algorithm which makes $Q$ best-response queries, simulating it against a no-regret learner may require $T = \exp(Q)$ rounds of play. For the special case of opponents with no-*adaptive*-regret algorithms (such as online gradient descent), we show that only $T = \mathrm{poly}(Q)$ rounds are required in the worst case. However, in general we show that exponential runtime can be necessary. In particular, we give an example of a game with $M$ actions where it is possible to learn the Stackelberg equilibrium in $\mathrm{poly}(M)$ rounds when playing against any no-swap-regret learning algorithm, but where it requires at least $\exp(M)$ rounds to learn this equilibrium when playing a mean-based no-regret algorithms.

## 1.2 Related work

The broader literature on no-regret learning in repeated games is substantial, covering many equilibrium convergence results varying assumptions. A recent line of work (5; 10; 21) considers problems related to optimizing one's reward when competing against a no-regret learner in a game. We extend these questions to consider the relationship and regret for an optimizer, as well as to settings where properties of the game are initially unknown, and give a series of separation results in terms of various notions of equilibrium. Also relevant is the literature on analysis of no-regret trajectory dynamics, and in particular (20) which shows a game in which no-regret dynamics outperform the reward of the Nash equilibrium. Additionally, there is also prior work considering regret minimization problems involving either best-responding or otherwise strategic agents (see e.g. (3; 18)), as well as work considering alternate regret notions or behavior models for repeated Stackelberg games (e.g. (12; 15)).

## 1.3 Notation and preliminaries

Throughout, we consider two-player bimatrix games $G = (A, B)$, where player $A$ ("the optimizer") has action set $\mathcal{A} = \{a_1, \ldots, a_M\}$ and player $B$ ("the learner") has action set $\mathcal{B} = \{b_1, \ldots, b_N\}$. When the optimizer plays action $a_i$ and the learner plays action $b_j$, the players receive rewards $u_A(a_i, b_j)$ and $u_B(a_i, b_j)$, respectively. We assume that the magnitude of each utility is bounded by a constant. The sets of mixed strategies for each player are denoted by $\Delta(\mathcal{A})$ and $\Delta(B)$, respectively;

when the optimizer plays a mixed strategy $\alpha \in \Delta(\mathcal{A})$ and the learner plays $\beta \in \Delta(\mathcal{B})$, the expected reward for the optimizer is given by $u_A(\alpha, \beta) = \sum_{i=1}^{M} \sum_{j=1}^{N} \alpha_i \beta_j u_A(a_i, b_j)$, with $u_B(\alpha, \beta)$ defined analogously. An action $b \in \mathcal{B}$ is a *best response* to a strategy $\alpha \in \Delta(\mathcal{A})$ if $b \in \arg\max_{b' \in \mathcal{B}} u_B(\alpha, b')$. We let $\mathsf{BR}(\alpha)$ be the set of all such actions for player $B$, and likewise $\mathsf{BR}(\beta)$ for player $A$.

## 2 Generalized equilibria and no-$\Phi$-regret learning

Here we introduce the notions of $\Phi$-regret and generalized equilibria, which we use to analyze the regret and reward of players in repeated bimatrix games under varying assumptions regarding the choice of regret benchmarks, the algorithms used, and the structure of the game.

Originally introduced in (14) and extended to general convex settings in (13), we consider the formulation of *linear* $\Phi$-regret as it relates to bimatrix games. Given a sequence of action pairs $(a_{i_1} b_{j_1}), \ldots, (a_{i_T} b_{j_T})$ for $T > 0$ and some set of functions $\Phi$, where each $f \in \Phi$ maps actions $\mathcal{A}$ to action profiles in $\Delta(\mathcal{A})$, we say that the $\Phi$-*regret* for the optimizer (and analogously for the learner) is

$$\mathsf{Reg}_\Phi(T) = \max_{f \in \Phi} \sum_{t=1}^{T} u_A(f(a_{i_t}), b_{j_t}) - u_A(a_{i_t}, b_{j_t}).$$

**Definition 1** (No-$\Phi$-regret learning). *We say a learning algorithm $\mathcal{L}$ is a* no-$\Phi$-regret *algorithm if, for some constant $c < 1$, we have that $\mathsf{Reg}_\Phi(\mathcal{L}, T) = O(T^c)$, where $\mathsf{Reg}_\Phi(\mathcal{L}, T)$ is the $\Phi$-regret corresponding to the action sequence played by $\mathcal{L}$.*

Some notable sets of regret comparator functions $\Phi$ are the constant maps $\mathcal{E}$ (corresponding to *external regret*), where all input actions are mapped to the same output action, and the "swap functions" $\mathcal{I}$ (corresponding to *internal regret*[1]), which contain all single swap maps $f_{ij} : [M] \to [M]$ where $f(i) = j$ and $f(i') = i'$ for $i' \neq i$. Imposing these constraints on players in a game results in a *(coarse) correlated equilibrium*, which are instances of our notion of *generalized equilibrium*.

**Definition 2** (Generalized $(\Phi_A, \Phi_B)$-equilibria). *A $(\Phi_A, \Phi_B)$-equilibrium $\varphi \in \Delta(\mathcal{A} \times \mathcal{B})$ in a two-player game is a joint distribution over action profiles $(a, b)$ such that player $A$ cannot increase their expected reward by deviating with some strategy in $\Phi_A$ and player $B$ cannot benefit by deviating with some strategy in $\Phi_B$.*

In contrast to the $\Phi$-equilibria considered by (14; 13), here we allow constraints to be asymmetric between players. While many equilibrium notions for two-player games impose symmetric regret constraints on each player (e.g. Nash, correlated, and coarse correlated equilibria), this need not always be the case. In Section 3, we highlight Stackelberg equilibria as a motivating example for considering more general notions of asymmetric equilibria from the perspective of $\Phi$-regret, to determine when one should deviate from simultaneous no-swap play, and in Section 4 we characterize the maximum reward attainable against no-regret learners in terms of asymmetric equilibria.

We say that the *value* of a game $G$ for player $A$ of a certain equilibrium class $(\Phi_A, \Phi_B)$, denoted $\mathsf{Val}_A(\Phi_A, \Phi_B)$ is the maximum reward obtainable by player $A$ at some $(\Phi_A, \Phi_B)$-equilibrium (with $\mathsf{Val}_B(\Phi_A, \Phi_B)$ defined symmetrically for player $B$). Likewise, we say that the *min-value* of a game for a player and an equilibrium class, denoted by e.g. $\mathsf{MinVal}_A(\Phi_A, \Phi_B)$ for player $A$, is the minimum reward for a player over all $(\Phi_A, \Phi_B)$-equilibria in a game. These capture the range of feasible average rewards under repeated play via $(\Phi_A, \Phi_B)$-regret dynamics.

**Proposition 1.** *For a repeated game over $T$ rounds where player $A$ uses a no-$\Phi_A$-regret algorithm and player $B$ uses a no-$\Phi_B$-regret algorithm, the average rewards obtained by each player are upper bounded by $\mathsf{Val}_A(\Phi_A, \Phi_B) + o(1)$ and $\mathsf{Val}_B(\Phi_A, \Phi_B) + o(1)$, respectively, and lower bounded by $\mathsf{MinVal}_A(\Phi_A, \Phi_B) - o(1)$ and $\mathsf{MinVal}_B(\Phi_A, \Phi_B) - o(1)$.*

We consider an $\varepsilon$-approximate $(\Phi_A, \Phi_B)$-equilibrium to be a joint profile distribution where each constraint is satisfied up to additive error $\varepsilon$, connecting Definitions 1 and 2 as follows.

**Proposition 2** (Convergence of no-$\Phi$-regret dynamics to generalized equilibrium). *Suppose after $T$ rounds of a game where player $A$ plays a no-$\Phi_A$-regret algorithm and player $B$ plays a no-$\Phi_B$-regret algorithm, player $A$ has average $\Phi_A$-regret $\leq \varepsilon$ and player $B$ has average $\Phi_B$-regret $\leq \varepsilon$. Let*

---

[1]We refer to internal and swap regret interchangeably, as our focus is primarily on rates with respect to $T$.

$\varphi^t := p_A^t \times p_B^t$ *denote the joint distribution over both players' actions at time $t$ and $\varphi := \frac{1}{T} \sum_{t=1}^{T} \varphi^t$* *denote the time-averaged history over joint player action distributions. Then, $\varphi$ is an $\varepsilon$-approximate* $(\Phi_A, \Phi_B)$*-equilibrium where*

$$\mathbb{E}_{(a,b) \sim \varphi} [u_A(a,b)] \geq \mathbb{E}_{(a,b) \sim \varphi} [u_A(f_A(a), b)] - \varepsilon, \text{ and}$$

$$\mathbb{E}_{(a,b) \sim \varphi} [u_B(a,b)] \geq \mathbb{E}_{(a,b) \sim \varphi} [u_B(a, f_B(b))] - \varepsilon$$

*for every possible deviation $f_A \in \Phi_A, f_B \in \Phi_B$. Likewise, if players $A$ and $B$ repeatedly play strategies corresponding to an $(\Phi_A, \Phi_B)$-equilibrium, then player $A$ is no-$\Phi_A$-regret and player $B$ is no-$\Phi_B$-regret.*

A general construction for no-$\Phi$-regret algorithms is given in (13), which immediately implies feasibility of dynamics which converge to *some* instance of any class of $(\Phi_A, \Phi_B)$-equilibria in a game, possibly requiring agents to use differing algorithms if constraints are asymmetric. We show that a much stronger claim is true: such a pair of algorithms exists for *any* $(\Phi_A, \Phi_B)$-equilibrium $\Psi$ in a game. These algorithms also satisfy a "best-of-both-worlds" property, in that they converge to $\Psi$ when played together, yet simultaneously maintain their corresponding regret guarantees against arbitrary adversaries.

**Theorem 1.** *Consider any game $G$. Suppose there exists a no-$\Phi_A$-regret learning algorithm $\mathcal{L}_A$ and a no-$\Phi_B$-regret learning algorithm $\mathcal{L}_B$. For any particular $(\Phi_A, \Phi_B)$-equilibrium $\Psi$ in a game $G$, there exists a pair of learning algorithms $(\mathcal{L}_A^*(\Psi), \mathcal{L}_B^*(\Psi))$ such that:*

- *The empirical sequence of play when Player $A$ uses $\mathcal{L}_A^*(\Psi)$ and Player $B$ uses $\mathcal{L}_B^*(\Psi)$ converges to $\Psi$.*

- *$\mathcal{L}_A^*(\Psi)$ and $\mathcal{L}_B^*(\Psi)$ are no-$\Phi_A$-regret and no-$\Phi_B$-regret, respectively, against arbitrary adversaries.*

Our approach is for the algorithms to initially implement a schedule of strategies which converges to $\Psi$. Yet, these algorithms also detect when their opponent disobeys the schedule by tracking their $\Phi$-regret with respect to $\Psi$, and after $o(T)$ violations can deviate indefinitely to playing a standalone no-$\Phi$-algorithm for all remaining rounds. Several of our results throughout make use of Theorem 1. Here we state a notable immediate implication for equilibrium selection.

**Corollary 1.1.** *For any equilibrium scoring function $\Gamma : \Delta(\mathcal{A} \times \mathcal{B}) \to \mathbb{R}$ with a unique optimum computable in finite time, there exists a pair of learning algorithms $(\mathcal{L}_A^*, \mathcal{L}_B^*)$ such that:*

- *The empirical distribution when player $A$ uses $\mathcal{L}_A^*$ and player $B$ uses $\mathcal{L}_B^*$ converges to $\text{argmax}_\Psi \Gamma(\Psi)$.*

- *$\mathcal{L}_A^*$ and $\mathcal{L}_B^*$ are no-$\Phi_A$-regret and no-$\Phi_B$-regret, respectively, against arbitrary adversaries.*

*Proof.* First optimize $\Gamma$ over $\Psi$ in finite time to find the unique optimum; then apply Theorem 1 to the resulting desired equilibrium. $\square$

Corollary 1.1 allows for optimizing for objectives such as total welfare or min-max utility for both players, and imposing conditions on generalized equilibria beyond $\Phi$-regret constraints (e.g. product constraints for Nash equilibria) by assigning arbitrarily low scores to invalid strategy profiles.

In subsequent sections, we will primarily focus the function classes $\mathcal{E}$ and $\mathcal{I}$ corresponding to external and internal regret as mentioned above, as the well empty set $\emptyset$ corresponding to unconstrained regret, and we additionally will consider the case when the game $G$ is initially unknown. Before continuing, we note that each player's values for any $(\Phi_A, \Phi_B)$-equilibrium class can be expressed via a linear program, whose size is polynomial in the game dimensions for these function classes of interest.

**Proposition 3.** *For any game $G$ and constraints $(\Phi_A, \Phi_B)$, both $\mathsf{Val}_A(\Phi_A, \Phi_B)$ and $\mathsf{Val}_B(\Phi_A, \Phi_B)$ are computable via linear programs with $MN$ variables and $\text{poly}(M, N, |\Phi_A|, |\Phi_B|)$ constraints. When $\Phi_A$ and $\Phi_B$ belong to $\{\emptyset, \mathcal{E}, \mathcal{I}\}$, the number of constraints is $\text{poly}(M, N)$.*

In general, these values for a player may differ under distinct notions of generalized equilibria; we give several examples of such concrete value separations in Appendix A (in Theorem 8). Our results in Section 3 illustrate a particularly stark separation of this form, in which it can often be *dominant* to deviate to a strategy where $\Phi$-regret constraints are violated.

# 3 Stability of no-swap-regret play

Here we address the following question: when is it the case that for two players in a game, it is an approximate (Nash) equilibrium for both players to play no-swap-regret strategies? More specifically, imagine a "metagame" where at the beginning of this repeated game, both players simultaneously announce and commit to a specific adaptive (and possibly randomized) algorithm they intend to run to select actions to play in the repeated game $G$ for the next $T$ rounds. In this metagame, for which games $G$ is it an $o(T)$-approximate Nash equilibrium for both players to play a no-swap-regret learning algorithm?

Of course, the answer to this question might depend on *which* specific no-swap-regret learning algorithm the agents declare. We therefore attempt to understand the following two questions:

- **Necessity:** For which games $G$ is it the case that there exists *some* pair of no-swap-regret algorithms which form a $o(T)$-approximate Nash equilibrium? (Equivalently, when is it *never* the case that playing no-swap-regret algorithms forms an approximate Nash equilibrium?)

- **Sufficiency:** For which games $G$ is it the case that *all* pairs of no-swap regret algorithms form $o(T)$-approximate Nash equilibria?

A central element of our analysis will be to consider the *Stackelberg equilibria* of a game.

**Definition 3** (Stackelberg Equilibria)**.** *The* Stackelberg equilibrium *of a game $G$ for player $A$ is the pair of strategies $(\alpha, b)$ given by* $\operatorname{argmax}_{\alpha \in \Delta(\mathcal{A}),\ b \in \mathsf{BR}(\alpha)} u_A(\alpha, b)$, *and the resulting expected utility for player $A$ is the* Stackelberg value *of the game, denoted* $\mathsf{Stack}_A$. $\mathsf{Stack}_B$ *is defined symmetrically.*

We can relate Stackelberg equilibria to our notions of generalized equilibria.

**Proposition 4.** *For any game $G$, we have that* $\mathsf{Stack}_A = \mathsf{Val}_A(\emptyset, \mathcal{I})$.

Here, any joint distribution over action profiles where player $B$ has zero swap regret constitutes a $(\emptyset, \mathcal{I})$-equilibrium for a game, and the optimal value for such an equilibrium for player $A$ coincides with the Stackelberg value. Further, each equilibrium set can be optimized over via a linear program.

Note that each value definition allows for tiebreaking in favor of player $A$. In general, simply playing the Stackelberg strategy $\alpha$ may not suffice to obtain $\mathsf{Stack}_A$ if the best response for Player $B$ is not unique. However, there are a number of mild conditions which are each sufficient to ensure the existence of an approximate Stackelberg strategy $\alpha'$ which yields a unique best response for player $B$ and obtains $\mathsf{Stack}_A - \varepsilon$ for any $\varepsilon > 0$. Here we consider a minimal such condition (essentially, no action is weakly dominated without also being strictly dominated).

**Assumption 1.** *In a game $G$, for each $b$, either $\mathsf{BR}(\alpha) = \{b\}$ for some $\alpha$, or $b \notin \mathsf{BR}(\alpha)$ for all $\alpha$. Likewise, for each $a$, either $\mathsf{BR}(\beta) = \{a\}$ for some $\beta$, or $a \notin \mathsf{BR}(\beta)$ for all $\beta$.*

We provide an efficient algorithmic procedure to answer both questions of necessity and sufficiency for a specific game $G$ satisfying Assumption 1. To do this, recall that when two players both employ no-swap regret strategies, they asymptotically (time-average) converge to some correlated equilibrium (here, corresponding to an $(\mathcal{I}, \mathcal{I})$-equilibrium). On the other hand, by defecting from playing a no-swap regret strategy (while the other player continues playing their no-swap regret strategy), a player can guarantee their Stackelberg value for the game. Moreover, as shown by (10), this is the *optimal* (up to $o(T)$ additive factors) best response to an opponent running a no-swap regret strategy. It thus suffices to understand how the utility a player might receive under a correlated equilibrium compares to the utility they receive under their Stackelberg strategy. For a fixed game $G$, let $\mathsf{Stack}_A = \mathsf{Val}_A(\emptyset, \mathcal{I})$ be the Stackelberg value for the first player, and $\mathsf{Stack}_B = \mathsf{Val}_B(\mathcal{I}, \emptyset)$ be the Stackelberg value for the second player. We have the following theorem.

**Theorem 2.** *Fix a game $G$ satisfying Assumption 1. The following two statements hold:*

1. *There exists some pair of no-swap-regret algorithms that form an $o(T)$-approximate Nash equilibrium in the metagame iff there exists a correlated equilibrium $\varphi$ in $G$ such that $u_A(\varphi) = \mathsf{Stack}_A$ and $u_B(\varphi) = \mathsf{Stack}_B$.*

2. *Any pair of no-swap-regret algorithms form an $o(T)$-approximate Nash equilibrium in the metagame iff for all correlated equilibria $\varphi$ in $G$, $u_A(\varphi) = \mathsf{Stack}_A$ and $u_B(\varphi) = \mathsf{Stack}_B$.*

*Moreover, given a game $G$, it is possible to efficiently (in polynomial time in the size of $G$) check whether each of the above cases holds.*

We obtain both claims by leveraging the construction in Theorem 1: best-case and worst-case correlated equilibria are feasible by some pair of no-swap-regret algorithms, and both players must simultaneously achieve close to their Stackelberg value for deviating to not be preferable. The characterization in Theorem 2 is algorithmically useful, but sheds little direct light on in which games or how often we would expect playing no-swap-regret to be an approximate equilibrium. It turns out that for many games, playing no-swap-regret is *not* an equilibrium; below we will show that for almost all games, if $G$ does not have a pure Nash equilibrium, at least one player has an incentive to deviate to their Stackelberg strategy.

**Definition 4.** *A property $P$ of a game holds for* almost all *games if, given any game $G$, property $P$ holds with probability $1$ for the game $G'$ formed by starting with $G$ and perturbing each of the entries $u_A(a_i, b_j)$ and $u_B(a_i, b_j)$ by independent uniform random variables in the range $[-\varepsilon, \varepsilon]$ (for any choice of $\varepsilon$). In other words, the property holds for almost all choices of the $2MN$ utility values that define a game (with respect to the standard measure on this space).*

**Theorem 3.** *For almost all games $G$, if $G$ does not have a pure Nash equilibrium, then there does not exist a pair of no-swap-regret algorithms which form a $o(T)$-approximate Nash equilibrium in the metagame for $G$.*

*Sketch.* We can show that if a correlated equilibrium has the same utility for a player as their Stackelberg value (a consequence of Theorem 2), then the correlated equilibrium must be a convex combination of valid Stackelberg equilibria. In almost all games, both players have unique Stackelberg equilibria (and Assumption 1 holds), which implies that this correlated equilibrium must actually be the Stackelberg strategy for both players simultaneously. This implies that it is a pure Nash equilibrium (since one action in a generic Stackelberg equilibrium is always pure). $\qquad\square$

Note that although Theorem 3 holds for almost all games, there are some important classes of games (most notably, zero-sum games) in the measure zero subset omitted by this theorem statement that both a) do not have pure Nash equilibria and b) have the property that playing no-swap-regret algorithms is an approximate equilibrium in the metagame (in particular, for zero-sum games, the Stackelberg value collapses to the value of the unique Nash equilibrium). Still, Theorem 3 shows that there are very wide classes of games for which playing no-swap-regret algorithms is not stable from the perspective of the agents.

Finally Theorem 3 requires that we deviate to our Stackelberg strategy, which may be hard to compute. One can ask whether there are games where efficient deviations – e.g., to algorithms with *weaker* regret guarantees – lead to strictly more utility for the deviating player. In Appendix B we show this is true in the following sense: there are games $G$ where $\mathsf{Val}_A(\mathcal{E}, \mathcal{I}) > \mathsf{MinVal}_A(\mathcal{E}, \mathcal{I}) = \mathsf{Val}_B(\mathcal{I}, \mathcal{I})$. That is, in such a game player $A$ can possibly strictly increase their utility by switching to a low-external-regret strategy, and such a switch will never decrease their utility.

## 4   Optimal rewards against no-regret learners

Here, we characterize the feasibility of optimizing one's reward against no-(external)-regret learners in terms of generalized equilibria. In contrast to the case of no-swap-regret learners, as shown by (10) there are games in which one can obtain $\Omega(T)$ more than the Stackelberg value over $T$ rounds against certain no-regret algorithms by playing an appropriate adaptive strategy. A major remaining open question from this line of work is determining the best feasible reward and corresponding optimal strategy against no-regret agents in arbitrary games. We resolve this question when considering the maximum over all possible no-regret algorithms: for any game, we can compute an upper bound on the feasible reward against any no-regret algorithm, and we show that there exists a specific no-regret algorithm against which we can obtain this reward via an efficiently implementable strategy.

**Theorem 4.** *For any game $G$, there exists a no-regret algorithm $\mathcal{L}$ and a strategy for player $A$ such that the total reward of player $A$ converges to $\mathsf{Val}_A(\emptyset, \mathcal{E}) \cdot T \pm o(T)$ when player $B$ uses $\mathcal{L}$.*

Here we again make use of the construction from Theorem 1. Note that by Proposition 1, this is optimal over all no-external-regret algorithms, as any adaptive strategy constitutes a no-$\emptyset$-regret

algorithm. By Proposition 3 we can identify the optimal $(\emptyset, \mathcal{E})$-equilibrium in poly$(M, N)$ time, which is sufficient to implement the algorithm $\mathcal{L}$ as well as our own strategy efficiently.

However, we additionally show that this bound is often unattainable against many standard no-external-regret algorithms. A property of many such algorithms (including Multiplicative Weights, Follow the Perturbed Leader, and Exp-3) is that they are *mean-based*, as formulated by (5).

**Definition 5** (Mean-based learning). *Let $\sigma_{i,t}$ be the cumulative reward resulting from playing action $i$ for the first $t$ rounds. An algorithm $\mathcal{L}$ is $\gamma$-mean-based if, whenever $\sigma_{i,t} \le \sigma_{j,t} - \gamma T$, the probability that the algorithm selects action $i$ in round $t + 1$ is at most $\gamma$, for some $\gamma = o(1)$.*

These algorithms resemble "smoothed" variants of Follow the Leader; they only play actions with probability higher than $o(1)$ if their cumulative reward thus far is not too far from optimal, and hence never play dominated strategies. However, in general, $(\emptyset, \mathcal{E})$-equilibria may contain *dominated* strategies, as is also the case for coarse correlated equilibria. This allows us to show the following.

**Theorem 5.** *Against any mean-based no-regret algorithm for player B, there are games where a $T$-round reward of $(\mathsf{Val}_A(\emptyset, \mathcal{I}) + \varepsilon) \cdot T$ cannot be reached by any adaptive strategy for player A, for any $\epsilon > 0$. However, for this same game, $\mathsf{Val}_A(\emptyset, \mathcal{E}) > \mathsf{Val}_A(\emptyset, \mathcal{I})$.*

In the appendix, we introduce a notion of "dominated-swapping external regret" which we use to characterize a class of games for which this holds, and we give a concrete example of such a game.

# 5 Learning Stackelberg equilibria in unknown games

Our results thus far have highlighted the primacy of the Stackelberg reward as an objective for repeated play against a learner: it is optimal against a no-swap learner and can sometimes be optimal against a mean-based learner, and it is almost always attainable against any learner. However, until now our strategies have assumed knowledge of the entire game, which may be unrealistic in many settings for which learning in games is relevant, particularly in terms of our opponent's rewards.

Here, we consider the challenge of learning the Stackelberg strategy via repeated play against a no-regret learner when only our own rewards are known, which is unaddressed in the literature to our knowledge; much of the prior work on learning Stackelberg equilibria assumes a *query* model, where one can observe the best response $\mathsf{BR}(\alpha)$ played by an opponent for any queried mixed strategy $\alpha$. While here we cannot immediately observe the best response of an opponent, as their actions are selected by a learning algorithm which may be slow to adapt to changes in our behavior, we give a reduction from query algorithms of this form to strategies for choosing our actions which enable us to *simulate* queries to $\mathsf{BR}(\alpha)$ against a learner, and we analyze the efficiency of this approach (in terms of rounds required for learning) under differing assumptions on the learner's algorithm.

For comparison of behavior across time horizons of varying lengths, it will be convenient for us to consider the notion of an *anytime* regret bound, which can be obtained from any base no-regret algorithm via doubling methods, as well as often via learning rate decay.

**Definition 6** (Anytime regret algorithms). *An algorithm is an* anytime no-$\Phi$-regret *algorithm if satisfies $\mathsf{Reg}_\Phi(t) = O(t^c)$ over the first $t$ rounds, for some $c < 1$ and any $t \le T$.*

We also recall the notion of adaptive regret; many no-external-regret algorithms such as Online Gradient Descent satisfy no-adaptive-regret bounds (see e.g. (25)).

**Definition 7** (Adaptive regret algorithms). *An algorithm $\mathcal{L}$ for player B is a no-adaptive-$\Phi$-regret algorithm if $\sup_{r,s\in[T]} \mathsf{Reg}_\Phi(\mathcal{L}, [r, s]) \le O(T^c)$, for some $c < 1$, where $\mathsf{Reg}_\Phi(\mathcal{L}, [r, s]) = \max_{f\in\Phi} \sum_{t=r}^{s} u_B(a_{i_t}, f(b_{j_t})) - u_B(a_{i_t}, b_{j_t})$.*

A key distinction between adaptive-regret algorithms like OGD and mean-based algorithms like FTPL is in in their "forgetfulness", and hence their ability to quickly adapt when rewards change. This has stark implications for the efficiency of learning Stackelberg equilibria, which we show can take *exponentially* longer against mean-based algorithms. As shown by (25), adaptive regret is closely connected with dynamic regret; we note that our results for adaptive-regret learners can also be extended to dynamic-regret learners.

## 5.1 Simulating query algorithms

Our approach will be to compute an $\varepsilon$-approximate Stackelberg strategy by simulating best response queries against a learner, after which point we can obtain an average reward approaching $\mathsf{Stack}_A - \varepsilon$ in each subsequent round, calibrating $\varepsilon$ in terms of $T$ as desired. The query complexity for such algorithms can depend on the geometry of the best response regions of the game, and unfortunately, as shown by (24), there are "hard" game instances which require exponentially many queries. This issue arises when the best response regions may be quite small but non-empty, as even finding a point in each region is information-theoretically difficult. We restrict our attention to games in which this does not occur, and fortunately efficient query do algorithms exist for this case.

**Assumption 2.** *For a game $G$ and any action $b_i$, we have that*

$$\Pr_{\alpha \sim \mathsf{Unif}(\Delta(M))}[b_i \in \mathsf{BR}(\alpha)] \in \{0\} \cup [1/\operatorname{poly}(\varepsilon^{-1}), 1],$$

*i.e. the volume of each $\mathsf{BR}$ region is either 0 or inverse polynomially large.*

**Proposition 5** ((19; 24)). *For a game $G$ satisfying Assumption 2, there is an algorithm which finds an $\varepsilon$-approximate Stackelberg strategy for player $A$ with $Q = \operatorname{poly}(M, N, 1/\varepsilon)$ queries to $\mathsf{BR}(\alpha)$.*

We note that while such algorithms can indeed obtain tighter approximation guarantees in terms of $\varepsilon$ (e.g. $O(\log(1/\varepsilon))$), the query complexity is still inverse polynomially related to the best response region volumes; we consider only $\varepsilon$-approximate equilibria due to challenges which are inherent to the no-regret learning setting, as the precision with which we can simulate a query is constrained by our time horizon. The key to our approach is to play according to a mixed strategy $\alpha$ until it saturates the relevant window of the learner's history, which induces them to play a best response. Against no-adaptive-regret learners, a best response will be induced quickly, as their regret is bounded even over small windows. However, for arbitrary no-regret learners, we have no promises other than the cumulative regret bound, which may require saturating the entire history for each query.

**Theorem 6.** *Suppose $\mathcal{E} \subseteq \Phi$. For a game satisfying Assumption 2, there is an algorithm which finds an $\varepsilon$-approximate Stackelberg strategy in $\operatorname{poly}(1/\varepsilon)^Q$ rounds against any anytime-no-$\Phi$-regret learner, and in $\operatorname{poly}(Q/\varepsilon)$ rounds against any no-adaptive-$\Phi$-regret calibrated for $T = \Theta(\operatorname{poly}(Q/\varepsilon))$, where $Q = \operatorname{poly}(M, N, 1/\varepsilon)$.*

## 5.2 Efficiency separations for mean-based and no-swap algorithms

We show here that the exponential dependence for mean-based algorithms is necessary: there exist games where learning the Stackelberg strategy *requires* exponentially many rounds against a particular mean-based algorithm. However, for the games we construct, we show that it is still possible to efficiently learn the Stackelberg strategy against a no-*swap*-regret learner.

**Theorem 7.** *There is a distribution over games $\mathcal{D}$ such that for a sampled game $G$:*

- *For any no-swap-regret learner used by the opponent, there is a strategy for the leader which yields an average reward of $\mathsf{Stack}_A - \varepsilon$ in $T = \operatorname{poly}(M/\varepsilon)$ rounds.*

- *There is a mean-based no-regret algorithm such that, when used by the opponent, there is no strategy for the leader which yields an average reward of $\mathsf{Stack}_A - \varepsilon$ over $T$ rounds unless $T = \exp(\Omega(M))$.*

Our construction includes a set of actions for player $B$ which are best responses to pure actions from player $A$, and one such pure strategy pair will necessarily constitute the Stackelberg equilibrium; identifying each best response suffices for player $A$ to identify the Stackelberg strategy. The game also includes a number of *safety* actions for player $B$, which yield no reward for player $A$ with any strategy, yet allow player $B$ to "hedge" between multiple actions of player $A$. This poses a barrier to optimizing against a mean-based learner: the history must be heavily concentrated on a single action to observe the best response, and as such the history length must grow by a constant factor for each observation. However, against a no-swap-regret learner, it suffices for the optimizer to only play each action for a polynomially long window in order to identify the learner's best response; we track the accumulation of a "swap-regret buffer" for any other action and show that it cannot be too large, limiting the number of rounds it can be played when it is not a current best response.

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

# A Properties and separations for generalized equilibria

## A.1 Proof of Proposition 1

*Proof.* The set of $(\Phi_A, \Phi_B)$-equilibria includes all strategy profile distributions in which both constraints are satisfied. If a player receives substantially more or less than the corresponding value, this would imply a violation of the regret constraints for at least one of the players' learning algorithms. □

## A.2 Proof of Proposition 2

*Proof.* The statement follows by observing that

$$\mathbb{E}_{(a,b)\sim\varphi}\left[u_{\{A,B\}}(a,b)\right] = \frac{1}{T}\sum_{t=1}^{T}\mathbb{E}_{(a,b)\sim\varphi^t}\left[u_{\{A,B\}}(a,b)\right]$$

$$\mathbb{E}_{(a,b)\sim\varphi}\left[u_A(f_A(a),b)\right] = \frac{1}{T}\sum_{t=1}^{T}\mathbb{E}_{(a,b)\sim\varphi^t}\left[u_A(f_A(a),b)\right]$$

$$\mathbb{E}_{(a,b)\sim\varphi}\left[u_B(a,f_B(b))\right] = \frac{1}{T}\sum_{t=1}^{T}\mathbb{E}_{(a,b)\sim\varphi^t}\left[u_B(a,f_B(b))\right]$$

which in turn are equivalent to the time-averaged utility of the play of players $A$ and $B$, the time-averaged utility for player $A$ under a deviation $f_A$, and the time-averaged utility for player $B$ under a deviation $f_B$. Applying the definition of average $\Phi$-regret and applying the given bounds on the $\Phi$-regret yields the conclusion of the first direction. The reverse direction follows by reversing the steps. □

## A.3 Proof of Proposition 4

*Proof.* Observe that under any strategy $(\alpha, b)$ where $b \in \mathsf{BR}(\alpha)$, player $B$ cannot have any swap-regret, and so any Stackelberg equilibrium is also a $(\emptyset, \mathcal{I})$-equilibrium. Further, the marginal distributions over the optimal $(\emptyset, \mathcal{I})$-equilibrium for player $A$ over each $b_i$ cannot have distinct expected value for player $A$, as otherwise this would contradict optimality, and so an optimal $(\emptyset, \mathcal{I})$-equilibrium is either a single Stackelberg equilibrium or a mixture of Stackelberg equilibria with equal value. □

## A.4 Proof of Proposition 3

*Proof.* By definition, the set of $(\Phi_A, \Phi_B)$-equilibria $\varphi$ is a sub-polytope of $\Delta(\mathcal{A} \times \mathcal{B})$ defined via the following linear constraints:

- For each $f_A \in \Phi_A$, we have that

$$\sum_{i\in[M]}\sum_{j\in[N]}\varphi_{ij}\,u_A(a_i,b_j) \geq \sum_{i\in[M]}\sum_{j\in[N]}\varphi_{ij}\,u_A(a_{f(i)},b_j).$$

- For each $f_B \in \Phi_B$, we have that

$$\sum_{i\in[M]}\sum_{j\in[N]}\varphi_{ij}\,u_B(a_i,b_j) \geq \sum_{i\in[M]}\sum_{j\in[N]}\varphi_{ij}\,u_B(a_{f(i)},b_j).$$

The value $\mathsf{Val}_A(\Phi_A, \Phi_B)$ corresponds to the element $\varphi$ of this polytope that maximizes $\sum_{i\in[M]}\sum_{j\in[N]}\varphi_{ij}\,u_A(a_i,b_j)$. Optimizing this linear function over the above polytope can be done in time $\mathrm{poly}(M, N, |\Phi_A|, |\Phi_B|)$ via any linear program solver. Computing $\mathsf{Val}_B(\Phi_A, \Phi_B)$ can be likewise done efficiently.

For player $A$, the regret comparator function sets $\emptyset$, $\mathcal{E}$, and $\mathcal{I}$ contain $0$, $M$, and $M^2$ elements respectively. In all three of these cases $|\Phi_A| = \mathrm{poly}(M)$; likewise, in all three of these cases $|\Phi_B| = \mathrm{poly}(N)$ (and thus we can efficiently compute these values when $\Phi_A, \Phi_B \in \{\emptyset, \mathcal{E}, \mathcal{I}\}$). $\quad\square$

### A.5 Reward separations

We show that with respect to optimal values, these equilibrium classes are often distinct, and there exist games where values do not collapse. The separations we show here consider the equilibrium cases either where both players have identical regret constraints, or where player $A$ is unconstrained. We note that while inspecting other cases, we identified similar examples for several other generalized equilibrium pairs, and we expect that strict separations exist between any distinct pair of generalized equilibria for the three regret notions we consider, in any direction not immediately precluded by the regret constraints. We are mostly interested in cases where $B$ is constrained, and $A$ may be constrained or unconstrained.

**Theorem 8.** *For each of the following, there exists a $4 \times 4$ game $G$ with rewards in $\{0, 1, 2\}$ where:*

1. $\mathsf{Val}_A\,(\emptyset, \mathcal{E}) > \mathsf{Val}_A\,(\emptyset, \mathcal{I}) > \mathsf{Val}_A\,(\mathcal{E}, \mathcal{E}) > \mathsf{Val}_A\,(\mathcal{I}, \mathcal{I})$

2. $\mathsf{Val}_A\,(\emptyset, \mathcal{E}) > \mathsf{Val}_A\,(\mathcal{E}, \mathcal{E}) > \mathsf{Val}_A\,(\emptyset, \mathcal{I}) > \mathsf{Val}_A\,(\mathcal{I}, \mathcal{I})$

*Proof.* We prove both results by exhibiting a game with the desired chain of inequalities, which we found by searching random examples of $4 \times 4$ games with values constrained in $\{0, 1, 2\}$ and computing the various values of the games with a linear programming library. The numerical values are easy to check with computation. The game $G_1 := (M_{A_1}, M_{B_1})$ satisfies the conditions for the first chain of inequalities, and the game $G_2 := (M_{A_2}, M_{B_2})$ satisfies the conditions for the second chain of inequalities. First we instantiate the game $G_1$:

$$
M_{A_1} := \begin{bmatrix} 1 & 0 & 0 & 0 \\ 1 & 0 & 0 & 1 \\ 2 & 2 & 0 & 2 \\ 0 & 2 & 0 & 0 \end{bmatrix}
\qquad
M_{B_1} := \begin{bmatrix} 0 & 2 & 0 & 0 \\ 1 & 1 & 1 & 0 \\ 1 & 0 & 2 & 0 \\ 1 & 0 & 0 & 1 \end{bmatrix}
$$

The corresponding values for game $G_1$ are simple to check:

1. $\mathsf{Val}_A\,(\emptyset, \mathcal{E}) = 8/5$.

2. $\mathsf{Val}_A\,(\emptyset, \mathcal{I}) = 4/3$.

3. $\mathsf{Val}_A\,(\mathcal{E}, \mathcal{E}) = 1$.

4. $\mathsf{Val}_A\,(\mathcal{I}, \mathcal{I}) = 0$.

Then we instantiate the game $G_2$:

$$
M_{A_2} := \begin{bmatrix} 2 & 0 & 1 & 0 \\ 2 & 1 & 1 & 0 \\ 0 & 2 & 1 & 2 \\ 2 & 0 & 2 & 1 \end{bmatrix}
\qquad
M_{B_2} := \begin{bmatrix} 1 & 0 & 1 & 2 \\ 0 & 1 & 2 & 0 \\ 1 & 0 & 2 & 0 \\ 0 & 2 & 1 & 1 \end{bmatrix}
$$

The corresponding values for game $G_2$ are simple to check:

1. $\mathsf{Val}_A\,(\emptyset, \mathcal{E}) = 13/7$.

2. $\mathsf{Val}_A\,(\mathcal{E}, \mathcal{E}) = 12/7$.

3. $\mathsf{Val}_A\,(\emptyset, \mathcal{I}) = 5/3$.

4. $\mathsf{Val}_A\,(\mathcal{I}, \mathcal{I}) = 4/3$.

$\quad\square$

## B  Deviation to weaker regret classes

In Section 3, we show that if two players are playing no-swap-regret strategies against one another, it is often in the interest of each player to switch to playing their Stackelberg strategy (in particular, this is true whenever the game does not have a pure Nash equilibrium). However, as we later argue, learning ones Stackelberg strategy in such a game can be difficult. It is therefore natural to ask whether there are beneficial deviations to computationally efficient strategies. In particular, is it ever in a player's interest to weaken their regret benchmark, and e.g. switch from playing a no-swap-regret strategy to a no-external-regret strategy?

We give an example showing this is true in a fairly strong sense: we exhibit a game $G$ where if player $A$ switches from playing a no-swap-regret algorithm to *any* no-external-regret algorithm, their asymptotic utility never decreases and sometimes strictly increases – i.e., there is no downside to switching to an external regret algorithm (and potentially a high upside). We have the following theorem.

**Theorem 9.** *There exists a game $G$ where* $\mathsf{MinVal}_A(\mathcal{E}, \mathcal{I}) \geq \mathsf{Val}_A(\mathcal{I}, \mathcal{I})$ *and* $\mathsf{Val}_A(\mathcal{E}, \mathcal{I}) \geq \mathsf{Val}_A(\mathcal{I}, \mathcal{I})$.

*Proof.* Consider the game $G$ specified by the two payoff matrices

$$M_A := \begin{bmatrix} 0 & 0 & 2 \\ 0 & 0 & 1 \\ 0 & 1 & 1 \end{bmatrix} \qquad M_B := \begin{bmatrix} 2 & 1 & 1 \\ 0 & 2 & 1 \\ 0 & 0 & 2 \end{bmatrix}.$$

The corresponding values for this game are simple to compute:

1. $\mathsf{Val}_A(\mathcal{I}, \mathcal{I}) = \mathsf{MinVal}_A(\mathcal{I}, \mathcal{I}) = 0$.

2. $\mathsf{MinVal}_A(\mathcal{E}, \mathcal{I}) = 0$.

3. $\mathsf{Val}_A(\mathcal{E}, \mathcal{I}) = 1$.

$\square$

## C  Proof of Theorem 1

*Proof.* Let $\varphi$ be the joint distribution over action pairs corresponding to $\Psi$. Let $T$ denote the total number of steps we run the algorithm for; we will use $t \leq T$ as a changing step size. Suppose both player $A$ and player $B$ know $\varphi$[2]. We will define $\mathcal{L}_A^*(\Psi)$ and $\mathcal{L}_B^*(\Psi)$ in two phases: in the first phase, $A$ and $B$ trust their opponent and play according to deterministic sequences corresponding to approximations of $\varphi$. If either player violates the other's trust $o(T)$ times, then the player defects to playing $\mathcal{L}_A$ or $\mathcal{L}_B$ respectively forever after.

First we elaborate upon the trusting phase. Both players consider windows of length $\mathsf{Length}(t)$ which is monotonically increasing in $t$ and also which grows sub-linearly in $t$. For concreteness, we pick a sub-linear monotonic increasing growth rate of $\mathcal{O}(\sqrt{t})$ and describe how to implement the schedule of window lengths. We can keep track of a real-valued variable $Z_t$ with $Z_1 = M \cdot N$, and after each window completes, update it by $Z_{t_{\text{next}}} = Z_t + \frac{1}{2\sqrt{t}}$ where $t$ is the step at the end of the window. To get an integral window length, we define $\mathsf{Length}(t) := \lfloor Z_t \rfloor$. Thus in this case, the $\mathsf{Length}(t)$ grows as $O(\sqrt{t})$, satisfying both conditions. Both players then compute a weighting instantiated with pairs of pure strategies by assigning $c_i := \lfloor \mathsf{Length}(t) \cdot \varphi_i \rfloor$ example pairs (each of weight $1/\mathsf{Length}(t)$) to pure strategy pair $i \in [M \cdot N]$. This weighted distribution approximates $\varphi$ given $\mathsf{Length}(t)$ samples. Note that the rounding approximation is feasible given only $\mathsf{Length}(t)$ samples since $\sum_{i=1}^{M \cdot N} c_i \leq \mathsf{Length}(t)$. These pure strategy pair samples are then lexicographically

---

[2]$\varphi$ can be communicated from Player $A$ to Player $B$ during a burn-in phase of length $> M \cdot N$, the dimension of the discrete joint distribution over pure player strategy pairs.

ordered. Then, both players act according to the pure strategies in order, thereby (over the window) achieving an $(M \cdot N)/\mathsf{Length}(t)$ $\ell_1$ approximation to $\varphi$:

$$\sum_{i=1}^{M \cdot N} \left| \varphi_i - \frac{c_i}{\mathsf{Length}(t)} \right| = \sum_{i=1}^{M \cdot N} \left| \varphi_i - \frac{\lfloor \mathsf{Length}(t) \cdot \varphi_i \rfloor}{\mathsf{Length}(t)} \right| \leq \frac{M \cdot N}{\mathsf{Length}(t)}.$$

This process repeats for every window.

The distrustful phase occurs if one of the players does not follow the agreed-upon instructions $T_{\text{distrust}}$ times, where $T_{\text{distrust}}$ is taken to be $o(T)$. After this many violations, Player $A$ defaults to playing $L_A$ and likewise Player $B$ defaults to playing $L_B$ ever after.

We now show that this algorithm satisfies both conditions in the theorem statement. First, if both players use $\mathcal{L}_A^*(\Psi)$ and $\mathcal{L}_B^*(\Psi)$, the play converges to $\varphi$, the joint distribution of play corresponding to $\Psi$. This point is immediate to observe since $(M \cdot N)/\mathsf{Length}(t) \to 0$ as $t \to \infty$ as $\mathsf{Length}(t)$ is monotone increasing in $t$.

Now we prove that both players are no-$\Phi$-regret with respect to any adversary. First we show no-$\Phi$-regret for both players in the case where Player $A$ plays $\mathcal{L}_A^*(\Psi)$ and Player $B$ plays $\mathcal{L}_B^*(\Psi)$. Let $\hat{\varphi}_t$ be the approximation to $\varphi$ implemented over the window corresponding to final step $t$, and suppose that $\|\varphi - \hat{\varphi}_t\|_1 < \varepsilon_t$. Recalling the proof of Theorem 1, for Player $A$ (and analogously for Player $B$) we can bound

$$\left| \mathbb{E}_{(a,b) \sim \varphi} [u_A(a,b)] - \mathbb{E}_{(a,b) \sim \hat{\varphi}_t} [u_A(a,b)] \right| = \left| (\varphi - \hat{\varphi}_t)^\top u_A \right|$$
$$\leq \|\varphi - \hat{\varphi}_t\|_1 \cdot \|u_A\|_2 \leq \varepsilon_t \cdot C \cdot \sqrt{M \cdot N},$$

where here we interpret $\varphi, \hat{\varphi}_t, u_A, u_B \in \mathbb{R}^{M \times N}$ as vectors over the space of all action pairs. Thus for this particular window, the overall gap from the expected reward for $\varphi$ is $\varepsilon_t \cdot C \cdot \sqrt{M \cdot N}$.

Then we can similarly upper bound $\mathbb{E}_{(a,b) \sim \hat{\varphi}_t} [u_A(f_A(a), b)] \leq \mathbb{E}_{(a,b) \sim \varphi} [u_A(f_A(a), b)] + \varepsilon_t \cdot C \cdot M\sqrt{N}$ for any choice of $f_A \in \Phi_A$:

$$(*) = \left| \mathbb{E}_{(a,b) \sim \varphi} [u_A(f_A(a), b)] - \mathbb{E}_{(a,b) \sim \hat{\varphi}_t} [u_A(f_A(a), b)] \right|$$
$$= \left| \sum_{k=1}^{M} \sum_{j=1}^{N} (\hat{\varphi}_t(k,j) - \varphi(k,j)) \cdot \sum_{i=1}^{M} f_A(a_k)_i \cdot u(\cdot, b_j) \right|$$
$$\leq \|\varphi - \hat{\varphi}_t\|_1 \cdot \| [f_A(a_1)^\top u_A(\cdot, b_1), \cdots, f_A(a_M)^\top u_A(\cdot, b_N)] \|_2$$
$$\leq \varepsilon_t \cdot \sqrt{M \cdot N} \cdot \max_{k,j} \|f_A(a_k)\|_2 \cdot \|u_A(\cdot, b_j)\|_2$$
$$\leq \varepsilon_t \cdot \sqrt{M \cdot N} \cdot 1 \cdot \sqrt{M \cdot C^2}$$
$$= \varepsilon_t \cdot M \cdot \sqrt{N} \cdot C.$$

Then recall that $\varepsilon_t \leq \frac{M \cdot N}{\mathsf{Length}(t)}$. Thus, overall, the average regret using due to the window is bounded by

$$\frac{1}{\mathsf{Length}(t)} \mathsf{Reg}_\Phi(\hat{\varphi}_t, t) \leq \frac{1}{\mathsf{Length}(t)} \mathsf{Reg}_\Phi(\varphi, t) + C_2 \cdot \frac{1}{\mathsf{Length}(t)},$$

where $C_2$ is another constant depending on $C, M, N$ and where we use the shorthand $\mathsf{Reg}_\Phi(\cdot, t)$ to denote the $\Phi$-regret over the window ending in step $t$. Now call $\hat{\varphi}$ the strategy where the joint distribution $\hat{\varphi}_t$ as previously defined gets played in each window $t$. Now we can bound the total $\Phi$-regret for $\hat{\varphi}$ by the sum of the $\Phi$-regrets for each window (maximizing $f_A \in \Phi_A$ over the steps in each window makes it more competitive than optimizing only one $f_A$ over the whole length $T$ sequence). Thus for total $\Phi$-regret, we have:

$$\mathsf{Reg}_\Phi(\hat{\varphi}, T) \leq \quad \mathsf{Reg}_\Phi(\varphi, T) + \mathsf{NumWindows}(T) \cdot C_2 \quad \leq \quad \mathsf{Reg}_\Phi(\varphi, T) + o(T),$$

where

$$\mathsf{NumWindows}(T) := \min_{\sum_{t=1}^{k} \mathsf{Length}(t) \geq T} k.$$

The last step follows since $\mathsf{NumWindows}(T) \leq o(T)$, because $\mathsf{Length}(T) \leq o(T)$.

Since we already know that the strategy $\varphi$ is no-$\Phi$-regret and $\mathsf{Length}(T)$ is $o(T)$, we have proven that playing $\hat{\varphi}$ is no-$\Phi$-regret in the case where Player $A$ plays $\mathcal{L}_A^*(\Psi)$ and Player $B$ plays $\mathcal{L}_B^*(\Psi)$.

The second case where the opposing player does not cooperate is easier: after at most $o(T)$ steps, the player switches to an algorithm $\mathcal{L}_A$ or $\mathcal{L}_B$ respectively which is no-$\Phi$-regret and incurrs only $o(T)$ additional regret. Thus the theorem statement holds.

$\square$

## D   Proof of Theorem 2

*Proof.* We begin with the first claim. To prove the forward direction, if there exists such a $\varphi$, then choose a pair of low-swap-regret algorithms $(\mathcal{L}_A, \mathcal{L}_B)$ such that the time-averaged trajectory over $T$ rounds is guaranteed to asymptotically converge to $\varphi$ (this is possible by either the results of (11), or our Theorem 1). That is, if the two players play strategy $\varphi_t$ at round $t \in [T]$, then $\hat{\varphi} = \frac{1}{T} \sum_t \varphi_t$ satisfies $||\hat{\varphi} - \varphi||_\infty = o(1)$. It follows that $\sum_t u_A(\varphi_t) \geq T \cdot u_A(\varphi) - o(T) = T \cdot \mathsf{Stack}_A -o(T)$ and therefore player $A$ has at most an $o(T)$ incentive to deviate (by (10), they can obtain at most $\mathsf{Stack}_A T + o(T)$ against $\mathcal{L}_B$). Symmetric logic holds for player $B$.

To prove the reverse direction, assume $\mathcal{L}_A$ and $\mathcal{L}_B$ are no-swap-regret algorithms such that $(\mathcal{L}_A, \mathcal{L}_B)$ is an $o(T)$-approximate Nash equilibrium in the metagame. Since they are no-swap-regret, the time-averaged play of these two algorithms for $T$ rounds must converge to an $o(1)$-approximate correlated equilibrium $\hat{\varphi}_T$; moreover, since $(\mathcal{L}_A, \mathcal{L}_B)$ is an $o(T)$-approximate Nash equilibrium, $\hat{\varphi}_T$ must have the property that $u_A(\hat{\varphi}_T) \geq \mathsf{Stack}_A -o(1)$ and $u_B(\hat{\varphi}_T) \geq \mathsf{Stack}_B -o(1)$. Taking the limit as $T \to \infty$ and selecting a convergent subsequence of the $\hat{\varphi}_T$, this shows there must exist a correlated equilibrium $\varphi$ with the desired properties.

Likewise, similar logic proves the second claim with the following modifications. In the forward direction, we can now choose any pair of low-swap-regret algorithms $(\mathcal{L}_A, \mathcal{L}_B)$, and any correlated equilibrium $\varphi$ they asymptotically converge to is guaranteed to have the property that $u_A(\varphi) = \mathsf{Stack}_A$ and $u_B(\varphi) = \mathsf{Stack}_B$. In the reverse direction, since any correlated equilibrium is implementable by some pair of low-regret algorithms (again, by Theorem 1), the same logic shows that all correlated equilibria $\varphi$ must satisfy $u_A(\varphi) = \mathsf{Stack}_A$ and $u_B(\varphi) = \mathsf{Stack}_B$.

Finally, to see that these two conditions are efficiently checkable, note that: i. the two values $\mathsf{Stack}_A$ and $\mathsf{Stack}_B$ are efficiently computable given the game $G$, and ii. the set of correlated equilibria $\varphi$ form a convex polytope defined by a small ($\mathsf{poly}(N, M)$) number of linear constraints (see Proposition 3). In particular, since $u_A(\varphi)$ and $u_B(\varphi)$ are simply linear functions of $\varphi$ for a given game $G$, we can efficiently check whether there exists any point in this polytope where $u_A(\varphi) = \mathsf{Stack}_A$ and $u_B(\varphi) = \mathsf{Stack}_B$. $\square$

## E   Proof of Theorem 3

*Proof.* We will show that (for almost all games $G$) if there is a correlated equilibrium $\varphi$ such that $u_A(\varphi) = \mathsf{Stack}_A$ and $u_B(\varphi) = \mathsf{Stack}_B$, then there exists a simultaneous unique Stackelberg equilibrium for both players in $G$, which must be a pure Nash equilibrium. Combined with Theorem 2, this implies the theorem statement.

We will rely on the following fact: in almost all games $G$, both players have a unique Stackelberg strategy. To see this, consider the following method for computing $A$'s Stackelberg strategy. For each pure strategy $b_j$ for player $B$, consider the convex set $A_j \subseteq \Delta(\mathcal{A})$ containing the mixed strategies for player $A$ which induce $b_j$ as a best response (i.e., $A_j = \{\alpha \in \Delta(\mathcal{A}) \mid b_j \in \mathsf{BR}(\alpha)\}$). Then, for each $j \in [N]$, compute the strategy $\alpha_j \in A_j$ which maximizes $u_A(\alpha_j, b_j)$. The Stackelberg value $\mathsf{Stack}_A$ is then given by $\max_j u_A(\alpha_j, b_j)$. In order for this to stem from a unique Stackelberg equilibrium, it is enough that: 1. the maximum utility is not attained by more than one $j$, and 2. for each $j$, the optimizer $\alpha_j \in A_j$ is unique.

These two properties are guaranteed to hold in almost all games. To see this, first note that the convex sets $A_j$ are determined entirely by the utilities $u_B$, so we will treat these as fixed. Now, given any convex set $A_j$, the extremal point in a randomly perturbed direction will be unique with probability 1 – but since $\alpha_j$ is simply the extremal point of $A_j$ in the direction specified by $u_A(\cdot, b_j)$ (which is a randomly perturbed direction), so $\alpha_j$ is unique in almost all games. Finally, if we perturb the magnitude of each of the utilities $u_A(\cdot, b_j)$ (keeping the direction the same), the maximizer $\max_j u_A(\alpha_j, b_j)$ will also be unique almost surely.

Let $(\alpha_A, b_A)$ be the Stackelberg equilibrium for player $A$ and let $(a_B, \beta_B)$ be the Stackelberg equilibrium for player $B$. Now, consider the aforementioned correlated equilibrium $\varphi \in \Delta(A \times B)$. We will begin by decomposing it into its marginals based on its first coordinate; that is, we will write $\varphi = \sum_{i=1}^{M} \lambda_i(a_i, \beta_i)$ for some mixed strategies $\beta_i \in \Delta(\mathcal{B})$ and weights $\lambda_i$ (with $\sum_i \lambda_i = 1$). By the definition of correlated equilibria, note that each $a_i$ belongs to $\mathsf{BR}(\beta_i)$. But this means that $u_B(a_i, \beta_i) \leq \mathsf{Stack}_B$, with equality holding iff $(a_i, \beta_i) = (a_B, \beta_B)$ (due to uniqueness of Stackelberg). Therefore, in order for $u_B(\varphi) = \mathsf{Stack}_B$, we must have that $\varphi = (a_B, \beta_B)$. By symmetry, we must also have that $\varphi = (\alpha_A, b_A)$. If both these are true, then $\varphi$ is a pure strategy correlated equilibrium of the game, and is hence a pure strategy Nash equilibrium (and moreover, is also the Stackelberg equilibrium for both $A$ and $B$). $\qquad\square$

## F   Proof of Theorem 4

*Proof.* By Theorem 1, there is a pair of $\emptyset$-regret and $\mathcal{E}$-regret algorithms $\mathcal{L}_A^*$ and $\mathcal{L}_B^*$ which converge to a $(\emptyset, \mathcal{E})$-equilibrium for which player $A$ obtains $\mathsf{Val}_A(\emptyset, \mathcal{E})$. By Proposition 1, this is optimal over all no-external-regret algorithms, as any adaptive strategy constitutes a no-$\emptyset$-regret algorithm. By Proposition 3 we can identify the optimal $(\emptyset, \mathcal{E})$-equilibrium in $\mathrm{poly}(M, N)$ time, which is sufficient to implement the algorithms $\mathcal{L}_A^*$ and $\mathcal{L}_B^*$ efficiently for any desired $T$. $\qquad\square$

## G   Dominated-swapping external regret bounds for mean-based algorithms

For the following proof (of Theorem 10), we introduce the following notion of *dominated-swapping external regret*, a tighter upper bound on the behavior of mean-based algorithms than the standard no-external-regret guarantee.

**Definition 8** (Dominated-swapping external regret). *For a game $G$, let $D(G)$ be the set of dominated strategies for player $B$, i.e. $b_i \in D(G)$ if $b_i \notin \mathsf{BR}(\alpha)$ for all $\alpha \in \Delta(M)$. For $j, k \in [N]$ define $g_{jk}(b_i)$ as:*

$$g_{jk}(b_i) = \begin{cases} b_j & b_i \notin D(G) \\ b_k & b_i \in D(G) \end{cases}$$

*i.e. $g_{jk}(b_i)$ swaps $b_i$ to $b_k$ if $b_i$ is dominated and plays $b_j$ otherwise. Let $\mathcal{E}_{D(G)} = \{g_{jk} : j, k \in [N]\}$ be the set of* dominated-swapping external regret *comparators.*

This definition leads to the following tighter upper bound on what is achievable against a mean-based no-regret algorithm.

**Theorem 10.** *For any game $G$ and any mean-based no-regret algorithm used by player $B$, there is no strategy for which the average reward of player $A$ converges to $\mathsf{Val}_A(\emptyset, \mathcal{E}_{D(G)}) + \varepsilon$, for any $\varepsilon > 0$.*

*Proof.* First, we observe that mean-based algorithms will never play a dominated strategy $b_i \in D(G)$ in more than $o(T)$ rounds. As $b_i$ is dominated, there is some $\delta > 0$ such that for every $\alpha \in \Delta(M)$, there is some $b_j$ where $u_B(\alpha, b_j) \geq u_B(\alpha, b_i) + \delta$. Let $\alpha_t$ denote the empirical distribution of player $A$'s actions up to time $t$. After some window of $O(\gamma T) = o(T)$ rounds we will have the cumulative rewards $\sigma_{i,t}$ and $\sigma_{j,t}$ satisfy $\sigma_{i,t} < \sigma_{j,t} - \delta t < \sigma_{j,t} - \gamma T$ under any $\alpha_t$ for some $b_j$ in each subsequent round, and so $b_i$ will never be played in more than $o(T)$ rounds.

We can also see that any such no-$\mathcal{E}$-regret algorithm is a no-$\mathcal{E}_{D(G)}$-regret algorithm. Suppose such an algorithm had $\mathcal{E}_{D(G)}$-regret $\epsilon T$, for $\epsilon > 0$; then, there is some $g_{jk}$ for which $U_B(\alpha_T, g_{jk}(\beta_T)) \geq U_B(\alpha_T, \beta_T) + \epsilon$. By the $\mathcal{E}$-regret guarantee this cannot occur if $j = k$, as any such function $g_{jj}$ is

equivalent to the fixed deviation rule for $b_j$. However, if this occurs for $j \neq k$, such an algorithm must have played dominated strategies in a total $\Omega(\epsilon T)$. This contradicts our assumption that no dominated strategy $b_i$ is played in more than $o(T)$ rounds, and so any mean-based no-$\mathcal{E}$-regret algorithm is also a no-$\mathcal{E}_{D(G)}$-regret algorithm, against which player $A$ cannot obtain average reward which converges to any amount higher than $\mathsf{Val}_A\left(\emptyset, \mathcal{E}_{D(G)}\right) + o(1)$. $\qquad\square$

## H  Proof of Theorem 5

|       | $b_1$ | $b_2$ | $b_3$ |
|-------|-------|-------|-------|
| $a_1$ | 1, 1  | 0, 0  | 3, 0  |
| $a_2$ | 0, 0  | 1, 1  | 0, 0  |

Figure 1: Game where $\mathsf{Val}_A\left(\emptyset, \mathcal{E}\right) > \mathsf{Val}_A\left(\emptyset, \mathcal{I}\right) = \mathsf{MBRew}_A$

*Proof.* Let $\mathsf{MBRew}_A$ denote the maximal reward obtainable by player $A$ when player $B$ uses a mean-based algorithm. Observe that $b_3$ is dominated for player $B$, and thus cannot be included in any $(\emptyset, \mathcal{I})$-equilibrium (by Theorem 10). Further, it will never be played by a mean-based learner for more than $o(T)$ rounds, as for any distribution over $a_1$ and $a_2$ the best response is either $b_1$ or $b_2$. As such, both $\mathsf{Val}_A\left(\emptyset, \mathcal{I}\right)$ and $\mathsf{MBRew}_A$ are at most $1 + o(1)$; a reward of $1 - o(1)$ is obtainable by committing to either $a_1$ or $a_2$ for each round. However, we can see that the optimal $(\emptyset, \mathcal{E})$-equilibrium $p$ for player $A$ includes positive mass on $(a_1, b_3)$, and yields an average reward of $\mathsf{Val}_A\left(\emptyset, \mathcal{E}\right) = 2$ for player $A$. Let $p_1$ be the probability on $(a_1, b_1)$, let $p_2$ be the probability on $(a_2, b_2)$, let $p_3$ be the probability on $(a_1, b_3)$, and let $p_0$ be the remaining probability. The reward for player $A$ is given by:

$$\mathsf{Rew}_A(p) = p_1 + p_2 + 3p_3$$

and $p$ defines a $(\emptyset, \mathcal{E})$-equilibrium if

$$\mathsf{Rew}_B(p) \geq \mathsf{Rew}_B(p \to b_i)$$

for each $b_i$, which holds if:

$$p_1 + p_2 \geq p_1 + p_3;$$
$$p_1 + p_2 \geq p_2;$$
$$p_1 + p_2 \geq 0.$$

Only the first constraint is non-trivial, and so the optimal $(\emptyset, \mathcal{E})$-equilibrium for player $A$ occurs by maximizing $p_1 + p_2 + 3p_3$ subject to $p_2 \geq p_3$, which yields a probability of 0.5 for both $p_2$ and $p_3$ (and 0 for $p_1$ and $p_0$), as well as an average reward of 2. As such, player $A$ cannot obtain a reward approaching $\mathsf{Val}_A(\emptyset, \mathcal{E})$, as their per-round reward is at most $1 + o(1)$. $\qquad\square$

## I  Proof of Theorem 6

*Proof.* We recall that the SU algorithm from (19) finds initial points $\alpha^*(b_i)$ in each best response region via random sampling, which takes takes $1/\operatorname{poly}(\varepsilon^{-1})$ queries in expectation. Then, upon calibrating for $O(\log(1/\varepsilon))$ bits of precision SU makes $\operatorname{poly}(M, N, \log(1/\varepsilon))$ queries, each of which can be taken to be a point on some grid of spacing $1/\operatorname{poly}(\varepsilon^{-1})$ within the simplex by the precision condition. The computed approximate Stackelberg strategy is then the optimal such point on the grid.

We first describe our strategy for simulating each query against an arbitrary anytime-no-regret learner; as $\mathcal{E} \subseteq \Phi$, we can restrict to considering only no-external-regret learners, as these regret constraints will always be satisfied. To implement a query $q$, greedily play the action whose historical frequency of play is the furthest below its target frequency in $q$. After $O(\operatorname{poly}(1/\varepsilon))$ rounds, the historical distribution will be within $1/\operatorname{poly}(\varepsilon^{-1})$ of $q$, and continuing the greedy selection strategy indefinitely will ensure that the history remains in a $1/\operatorname{poly}(\varepsilon^{-1})$-ball around $q$. Let $t_q$ be the time at which this occurs. After maintaining the greedy strategy for $q$ for an additional $\omega(t_q^c)$ rounds, the anytime regret bound ensures that most frequently played item must indeed be the best response response to some point in the ball around $q$, provided that this ball is contained entirely inside some best response region $R_j$. For the sampling step, a taking sufficiently small grid (but still $1/\operatorname{poly}(\varepsilon^{-1})$) ensures that random sampling still suffices to find a point a point in each best response region even if our

queries may be adversarially perturbed to neighboring points on the grid, as each region is convex and has volume at least $1/\operatorname{poly}(\varepsilon^{-1})$. To address the issue for the line search steps, it suffices to take an additional step along each search conducted by $\mathsf{SU}$ before termination, where we then take each hyperplane boundary estimate to be one step inward along the grid from where our search terminates, maintaining a buffer between each hyperplane estimate in which all our points of uncertainty must lie. This adds at most a constant factor to our query complexity, and impacts our approximation by $1/\operatorname{poly}(\varepsilon^{-1})$, which then yields us a runtime of $\operatorname{poly}(1/\varepsilon)^Q$ rounds.

For the case of a no-adaptive-regret learner, suppose such an algorithm is calibrated for $T = O(Q^{C_1}(1/\varepsilon)^{C_2})$; then, over any window of length $W$ its regret is at most $O\left((Q^{C_1}(1/\varepsilon)^{C_2})^c W^{-1}\right)$. Taking $W = \omega((Q^{C_1}(1/\varepsilon)^{C_2})^c)$ yields a per-round regret of at most $o(1)$ over the window, and so an algorithm must play a best response in $W - o(W)$ of the rounds. For sufficiently large $C_1$ and $C_2$, each $W$ is large enough to yield the same precision we required for the anytime case, where now we greedily play the action whose frequency is furthest below its target *since our previous query terminated*, which allows us to again simulate the $O(Q)$ queries in $\operatorname{poly}(Q/\varepsilon)$ rounds (accounting for the robustness checks) while yielding $\Theta(WQ) = o(T)$. $\qquad\square$

## J  Proof of Theorem 7

*Proof.* Our game consists of $M$ actions $\mathcal{A}$ for the optimizer, and $N = 2M + \binom{M}{2}$ actions for the learner, which are divided into $M$ *primary* actions $\mathcal{B}$, $M$ *secondary* actions $\mathcal{S}$, and $\binom{M}{2}$ *safety* actions $\mathcal{Y}$.

If we restrict the learner to only playing primary actions, the game somewhat resembles a coordination game, where each pure strategy pair $(a_j, b_j)$ is a Nash equilibrium. However, the set $\mathcal{B}$ is comprised of both *undominated* actions $\mathcal{B}_U$ and *dominated* actions $\mathcal{B}_D$, which are unknown to the optimizer, and where each $b_j \in \mathcal{B}_d$ is weakly dominated by the secondary action $s_j$. The optimizer receives reward 0 whenever the learner plays a secondary action, and so the challenge for the optimizer is to identify the pair $(a_j, b_j)$ which maximizes $u_A(a_j, b_j)$, for $b_j \in \mathcal{B}_D$, which will be the Stackelberg equilibrium. Further, the safety actions $y_{ij}$ essentially allow the learner to hedge between two actions; this does not pose substantial difficulty for the optimizer when the learner is no-swap-regret, yet creates an insurmountable barrier for learning the Stackelberg equilibrium in sub-exponential time against a mean-based learner.

An instance of a game $G \in \mathcal{G}$ is specified by the partition of $\mathcal{B}$ into $\mathcal{B}_U$ and $\mathcal{B}_D$. There is an action $s_j \in \mathcal{S}$ for each $j$, and for each pair $(i, j)$ with $i < j$ there is an action $y_{ij} \in \mathcal{Y}$. The rewards for a game $G$ are as follows. For any strategy pair, the optimizer's utility is given by:

- $u_A(a_j, b_j) = j/M$ for $b_j \in \mathcal{B}$;
- $u_A(a_i, b_j) = 0$ for $b_j \in \mathcal{B}$ and with $i \neq j$;
- $u_A(a_i, s_j) = 0$ for any $s_j \in \mathcal{S}$;
- $u_A(a_i, y_{jk}) = 0$ for any $y_{jk} \in \mathcal{Y}$;

and the learner's utility is given by:

- For $b_j \in \mathcal{B}_U$:
    - $u_B(a_j, b_j) = 1$;
    - $u_B(a_i, b_j) = 0$ for $i \neq j$;
- For $b_j \in \mathcal{B}_D$:
    - $u_B(a_i, b_j) = 0$ for any $i$;
- For $s_j \in \mathcal{S}$:
    - $u_B(a_j, s_j) = 1$ if $b_j \in \mathcal{B}_D$;
    - $u_B(a_j, s_j) = 0$ if $b_j \in \mathcal{B}_U$;

- $u_B(a_i, s_j) = 0$ for $i \neq j$;

- For $y_{ij} \in \mathcal{Y}$:

    - $u_B(a_i, y_{ij}) = u_B(a_j, y_{ij}) = 2/3$;
    - $u_B(a_k, y_{ij}) = 0$ for $i, j \neq k$.

We assume that $\mathcal{B}_U$ is non-empty, and so there is some optimal pure Nash equilibrium $(a_i^*, b_i^*)$ which yields a reward of $i/M$; it is simple to check that this is also the Stackelberg equilibrium.

**Optimizing against no-swap learners.** First, we give a method for matching the Stackelberg value against an arbitrary no-swap-regret learner, which corresponds to the pair $(a_j, b_j)$ for the largest value $j$ such that $b_j \in \mathcal{B}_U$. Consider a no-swap-regret learner which obtains a regret bound of $\tau = O(T^c)$ over $T$ rounds. Let $\mathsf{SR}_t(b, b')$ for any learner actions $b$ and $b'$ denote the $t$-round cumulative swap regret between $b$ and $b'$, i.e. the total change in reward which would have occurred if $b'$ was played instead for each of the first $t$ rounds in which $b$ was played. To model the behavior of an arbitrary no-swap-regret learner, we disallow the learner from taking any action which would increase $\mathsf{SR}_t(b, b')$ above $\tau$, given the loss function for the current round, and otherwise allow the action to be chosen adversarially. While our model is deterministic for simplicity, it is straightforward to extend to the analysis to algorithms whose regret bounds hold in only expectation, e.g. by considering a distribution over values of $\tau$ in accordance with Markov's inequality (as no algorithm can have negative expected regret against arbitrary adversaries) and considering our expected regret to the Stackelberg value.

Our strategy for the optimizer is:

- For each $i \in [M]$, play $a_i$ until either $b_i$ or $s_i$ is observed at least $t^* > \tau$ times;

- Return $a_i^*$ for the largest $i$ such that $b_i$ is observed $t^*$ times.

We show that this takes at most $O(T^c \cdot M^3)$ rounds. Once $a_i^*$ is identified, we can commit to playing it indefinitely, at which point the learner must play $b_i^*$ in all but at most $O(T^c \cdot \mathrm{poly}(M))$ rounds, and so with $T = O(\mathrm{poly}(M/\varepsilon))$ rounds we can increase the total fraction of rounds in which $(a_i^*, b_i^*)$ is played to $1 - \varepsilon$, which yields the desired average reward bound.

The key to analyzing the runtime of our strategy is to consider the "buffer" in regret between any pair of actions before the threshold of $\tau$ is reached, which enables us to the bound the number of rounds in which instantaneously suboptimal actions are played. Note that prior the start of window $i$ (where $a_i$ is played), both $b_i$ and $s_i$ obtain reward 0 in each round, and as such cannot decrease their expected regret relative to any other action, as all rewards in the game are non-negative. Further, for any previous window $j$, both $b_i$ and $s_i$ incur regret of 1 with respect to either $b_j$ or $s_j$, as well as between the suboptimal and optimal action in window $i$, and thus cannot be observed more than $\tau$ times in the window. As such, observing $b_i$ at least $t^*$ times in window $i$ indicates that $b_i \in \mathcal{B}_U$ (and likewise observing $b_i$ at least $t^*$ times indicates that $b_i \in \mathcal{B}_D$).

Any action $b \neq \mathsf{BR}(a_i)$ will incur positive swap regret with respect to $\mathsf{BR}(a_i)$, and cannot be played in window $i$ once $\mathsf{SR}_t(b, \mathsf{BR}(a_i)) \geq \tau$. Each action begins with $\mathsf{SR}_1(b, \mathsf{BR}(a_i)) = 0$ at time $t = 1$; for each of the learner's actions, we consider the rate at which its buffer decays, as well as instances in which swap regret can decrease:

- Previously optimal $b \in \mathcal{B} \cup \mathcal{S} \setminus \mathsf{BR}(a_i)$: actions in $\mathcal{B} \cup \mathcal{S}$ can only accumulate negative swap regret with respect to $\mathsf{BR}(a_i)$ during rounds in which they were previously optimal; any previous optimum $b = \mathsf{BR}(a_j)$ for $j < i$ was played at most $t^*$ times during window $j$, and so we have that $\mathsf{SR}_t(b, \mathsf{BR}(a_i)) \geq -t^*$.

- All $b \in \mathcal{B} \cup \mathcal{S} \setminus \mathsf{BR}(a_i)$: ignoring any previously accumulated regret buffer, each of these $2M - 1$ actions can be played at most $\tau$ rounds during window $i$ before exhausting their initial buffer. Accounting for possible previous optima with $\mathsf{SR}_t(b, \mathsf{BR}(a_i)) < 0$, the number of rounds during window $i$ in which some $b \in \mathcal{B} \cup \mathcal{S} \setminus \mathsf{BR}(a_i)$ is played is at most $Mt^* + (2M - 1)\tau$.

- Safety actions $y_{jk} \in \mathcal{Y}$: Suppose neither $a_j$ or $a_k$ have been played yet by the optimizer, including in the current window. As was the case for other actions which have never yielded positive instantaneous reward, $y_{jk}$ can be played at most $\tau$ times before $\mathsf{SR}_t(y_{jk}, \mathsf{BR}(a_i)) \geq \tau$. If $j = i$, i.e. this is the first window in which $y_{jk}$ obtains positive instantaneous reward, the per-round regret is $1/3$, and so at it can be played for most $3\tau$ rounds. Further, $y_{jk}$ a obtains a regret of $-2/3$ with respect to $\mathsf{BR}(a_k)$. If $k = i$ and the window for $a_j$ has already been completed, $y_{jk}$ can be played for at most $9\tau$ rounds, as initially we have that $\mathsf{SR}_t(y_{jk}, \mathsf{BR}(a_i)) \geq -2\tau$, which again increases by $1/3$ per round. We then have that the total amount of rounds with safety actions played during window $i$ is at most $(12M + M^2)\tau$, as there are fewer than $M^2$ total safety actions, and fewer than $M$ in each of the latter cases.

This yields a per-window runtime across all actions of at most $Mt^* + (M^2 + 10M - 1)\tau$, which is $O(T^c \cdot M^3)$ across all windows, and so we obtain the desired result for optimizing against arbitrary no-swap-regret learners.

**Optimizing against mean-based learners.** Here, we show that there are mean-based no-regret algorithms for which exponentially many rounds are required for an optimizer to approximate the Stackelberg value against a learner. When considering horizons which are superpolynomial in the parameters of the game, it is most natural to consider algorithms with regret bounds which are non-trivial for smaller horizons, as well as an anytime variant of the mean-based property. We define an extension of the classical Multiplicative Weight Updates algorithm (MWU; see (2) for a survey), called Rounded Mean-Based Doubling, which inherits both properties in the anytime setting.

---
**Algorithm 1** Rounded Mean-Based Doubling (RMBD)
___
Initialize and run MWU for $T_1 := 2$ rounds and $n$ actions.
Let $T_2 := 2T_1$ and $i := 2$.
**while** $T_i \leq T$ **do**
  Initialize MWU for $T_i$ rounds and $n$ actions.
  Simulate running MWU for $T_{i-1}$ rounds, using the average of the first $T_{i-1}$ rewards each round.

  For $T_{i-1}$ rounds, run MWU with action probabilities rounded to multiples of $4\gamma = \tilde{O}(T_i^{-1/2})$.
  Let $T_{i+1} = 2T_i$ and $i := i + 1$.
**end while**

---

**Lemma 6.** *When running* RMBD *for $T$ rounds, the following hold at any round $t \leq T$:*

- RMBD *has cumulative regret $\tilde{O}(n\sqrt{t})$;*

- *If action $j$ has the highest cumulative reward and $\sigma_{i,t} \leq \sigma_{j,t} - \tilde{O}(\sqrt{t})$, then action $i$ is played with probability 0 at round $t$.*

*Proof.* Let $C\sqrt{t}$ bound the regret of MWU over $t$ rounds (where $C = O(\sqrt{\log n})$), and let $D = \sqrt{2}C + \tilde{O}(n)$. We can bound the regret of RMBD over $T_i$ rounds by $D\sqrt{T_i}$ via induction (which holds trivially at $T_1$). Suppose it holds for some $T_i$. Let $R(T_i)$ be the true reward obtained by RMBD over $T_i$ rounds, which is at least $\sigma_{j^*,T_i} - D\sqrt{T_i}$, where $\sigma_{j^*,T_i}$ is the cumulative reward of the best action over $T_i$ rounds. Consider our simulation of MWU over $T_i$ rounds using the average reward function. As the reward function is identical each round, and the cumulative reward for each action $j$ is equivalent under averaging, the measured reward $\hat{R}(T_i)$ from the simulated run is at most $\sigma_{j^*,T_i}$ after $T_i$ rounds. Upon continuing to run this instance of MWU for an additional $T_i$ rounds, the regret bound ensures that the total measured reward $\hat{R}(T_{i+1})$ is at least $\sigma_{j^*,2T_i} - C\sqrt{2T_i}$. Rounding probabilities contributes at most an additional $2n\gamma T_i$ to the regret; it suffices to implement rounding by reallocating probability mass from any $p_{i,t} < 2\gamma$ onto other actions arbitrarily, to avoid renormalization. The total reward of RMBD over $2T_i = T_{i+1}$ is given by its cumulative reward at $T_i$, as well as the additional reward obtained by the MWU instance over the next $T_i$ rounds, and so we

have that

$$R(T_{i+1}) = R(T_i) + \hat{R}(T_{i+1}) - \hat{R}(T_i)$$
$$\geq \sigma_{j^*,T_{i+1}} - D\sqrt{T_i} - C\sqrt{2T_i} - 2n\gamma T_i$$
$$\geq \sigma_{j^*,T_{i+1}} - D\sqrt{T_{i+1}},$$

which yields the bound for every $T_i$. We can extend this to any $t \in [T_i, T_{i+1}]$ with at most a factor 2 increase to cumulative regret.

To bound the selection frequency of actions with suboptimal cumulative reward, we recall the mean-based analysis of MWU given in Theorem D.1 from (5), which shows that the selection frequency $p_{k,t}$ for action $k$ at time $t$ is at most $\gamma = \frac{2\log(\sqrt{T\log n})}{\sqrt{T\log n}}$ if $\sigma_{k,t} \leq \sigma_{j,t} - \gamma T$ for the action $j$ with highest cumulative reward. As such, any action whose cumulative reward $\sigma_{k,t} \leq \sigma_{j,t} - \tilde{O}(\sqrt{t})$ will be played with probability 0. $\qquad\square$

Suppose a learner plays the action with highest cumulative reward at each round for $t_{\text{burn}} = \tilde{\Omega}(M^2)$ rounds, then plays RMBD thereafter for a total of $T$ rounds. Note that this maintains the both properties of RMBD for all $t$. We show that at least $T = \exp(\Omega(M))$ rounds are required to identify the Stackelberg strategy. The optimizer must check the learner's pure best response to each $a_j$ for identification with certainty, and it is straightforward to construct a distribution in which any strategy which does not observe $\mathsf{BR}(a_j)$ for all $j$ will have linear regret to $\mathsf{Stack}_A$ in expectation (e.g. where $\mathcal{B}_U$ contains one action chosen uniformly at random). The difficulty in exploration of the best responses comes from the safety actions, as $a_j$ must have been played more frequently than any other action in order to not be dominated by some safety action. Let $\rho_{j,t}$ denote the number of rounds in which the optimizer has played $a_j$ out of the first $t$. Observe that by construction of the game and the properties of RMBD, an primary or secondary action $b_j$ or $s_j$ in $\mathsf{BR}(a_j)$ will only be played with positive probability when:

$$\rho_{j,t} \geq \frac{2}{3}(\rho_{j,t} + \rho_{k,t}) - \tilde{O}(\sqrt{t})$$
$$= 2\rho_{k,t} - \tilde{O}(\sqrt{t})$$

for all $k$, which necessitates that $\rho_{j,t} \geq \frac{2t}{M} - \tilde{O}(\sqrt{t})$. Taking $t_{\text{burn}}$ sufficiently large, we have that $\rho_{j,t} \geq \frac{3}{2}\rho_{k,t}$ for any $t \geq t_{\text{burn}}$ and all $k$. For any subsequent observation $\mathsf{BR}(a_k)$ at $t'$, we must have that $\rho_{k,t'} \geq \frac{3}{2}\rho_{j,t}$, and so the number of rounds required to play an action before observing its best response grow at a rate of at least $(3/2)^M$, which completes the proof.

$\qquad\square$

