# OpenReview forum: "Is Learning in Games Good for the Learners?"
_NeurIPS.cc/2023/Conference — NeurIPS 2023 spotlight_

### Official Review · Reviewer_VEjN · 2023-06-30

**Soundness:** 3 good
**Presentation:** 3 good
**Contribution:** 3 good
**Rating:** 7
**Confidence:** 2

**Summary:**

The paper studies several trade-offs that existed in learning in games. The first is the tradeoff between regret and rewards, for which a generalized notion of equilibria is introduced. It is then investigated whether running a no-swap regret learning algorithm is efficient. It is shown that this depends on the form of the games, in some games running a no-swap regret learning algorithm can be more efficient than employing the Stackelberg strategy. The same question is investigated for learning in game against a no-external regret learning algorithm.

**Strengths:**

Overall, the paper is really well-written and well-organized. The problem investigated and the results presented are interesting. The generalized notion of equilibria seems to be useful for other analyses of learning in games. It is also important to study the performance of different types of algorithms in games, (such as mean-based no-regret learning algorithms), these results can lead to further insights into algorithm designs for learning in games.

**Weaknesses:**

Overall the paper is pretty solid, a rather minor weakness is that the presentation of the paper can still be improved.
There are quite a number of results presented in the paper. As a result, section 1.1 is a really lengthy section. It seems like the authors are trying to summarize the question investigated, related works, and the obtained results in this section. This can lead to some confusion as some of the notions are yet to be introduced in the paper and this is still at the very beginning of the paper. I would suggest making the section more concise and putting some of the discussion in the later parts of the paper.

**Questions:**

1. Could the authors elaborate on why exponential weights, FTPL etc algorithms are mean-based? Also, Theorem 4 seems to be stated with respect to average reward, while Theorem 5 seems to be saying that a mean-based algorithm cannot attain the total rewards (which seems to be not surprising?), I wonder how these two Theorems should be interpreted together.

2. It is mentioned that Proposition 5 can be improved, though the query complexity is still inversely proportional to the best response region volume. But from Theorem 6, it seems that through stimulating the best response queries, the complexity is independent of the best response region volume.

3. It seems to be that mean-based algorithms often have no external-regret. I wonder if they can be also no swap regret? If so, how should one interpretate Theorem 7?

---

> ### Author Rebuttal · Authors · 2023-08-04
>
> Thank you for your comments! We will certainly take another pass on clarifying the narrative in the introduction, and can move some of the discussion of results to later in the paper.
>
> In terms of mean-based algorithms, all of these algorithms resemble approximate/smoothed versions of Follow the Leader, in that actions are almost never chosen unless they are close to historically optimal (see Appendix D of [BMSW17] for a more thorough analysis).
>
> We apologize for the confusion regarding Theorems 4 and 5; both of these theorems can be interpreted either in terms of total or average reward (by multiplying/dividing by $T$), and we will change the Theorem 5 statement for consistency. The takeaway for Theorem 5 is that all mean-based algorithms are strictly stronger than “worst-case” no-regret algorithms for certain games, and thus are harder to exploit (and an average reward of $\text{Val}_A (\emptyset, \mathcal{E})$ cannot be approached).
>
> As for Proposition 5, the exact BR query complexity for Stackelberg equilibria depends on a number of different parameters related to the structure of the game, and there are some technical intricacies we avoid discussing in the body for simplicity; to summarize, all algorithms which obtain accuracy $\epsilon$ with $\text{poly}(1/\epsilon)$ queries require that the best response regions have volume at least $1/\text{poly}(1/\epsilon)$. However, in the query model the accuracy of these algorithms can be boosted to e.g. $\log(1/\epsilon)$ (by binary searching), yet the relationship between query complexity and volume is still fixed (finding initial feasible points is the bottleneck). We are unable to take advantage of this accuracy boost in our setting due to the imprecision inherent in the learning setting. As such, we make use of the weaker bound so as to align with the error terms resulting from learning. Once the query complexity is established to be $\text{poly}(1/\epsilon)$, Theorem 6 only requires that the volume is at least $1/\text{exp}(1/\epsilon)$, but we still need Assumption 2 nonetheless to establish the query complexity. We will clarify this discussion in the paper.
>
>
> Finally, while many mean-based algorithms are no-(external)-regret, results from [DSS19] imply that no mean-based algorithms can be no-swap-regret (or no-internal-regret) when there are more than two actions (via an example of a 3-action game where the maximum reward against any no-swap learner is strictly below that obtainable against any mean-based learner). Note that the definitions of external and internal regret coincide for only 2 actions.
>
>
> [BMSW17] Selling to a No-Regret Buyer - Braverman, Mao, Schneider, Weinberg
> [DSS19] Strategizing against No-regret Learners - Deng, Schneider, Subramanian

---

### Official Review · Reviewer_vLcT · 2023-07-01

**Soundness:** 4 excellent
**Presentation:** 2 fair
**Contribution:** 3 good
**Rating:** 7
**Confidence:** 4

**Summary:**

This submission studies questions surrounding playing against a no-regret learner in a repeated game setting. These questions are motivated by previous observations that while it is known that when all players play no-regret strategies the empirical frequency of play approaches an equilibria, a player can sometimes do better by deviating to a different strategy/algorithm (that is not no-regret).

In particular, the authors focus on four questions: (1) When does reward trade off with regret? (2) Under what game settings is playing a no-swap-regret algorithm a stable equilibrium? (3) How to play against a no-regret learner? (4) How can one learn the Stackelberg strategy through repeated play against a no-regret learner?

Towards answering question (1), the authors consider a generalized notation of equilibrium, (Phi-A, Phi-B)-equilibrium, where player A is restricted to playing no-Phi-A-regret strategies and player B is restricted to playing no-Phi-B-regret strategies. Each choice of (Phi-A, Phi-B) induces a polytope of (Phi-A, Phi-B)-equilibria. They show that for any equilibria in this polytope, there exists a pair of no-regret algorithms in (Phi-A, Phi-B) which converge to it.

To assist in answering question (2), the authors consider a "metagame", in which at the beginning of a repeated game, both players simultaneously announce and commit to an algorithm to use during the game. The authors main result towards answering (2) are a set of sufficient and necessary conditions for (a) some pair of no-swap-regret algorithms to form a Nash equilibrium in the metagame and (b) all pairs of no-swap regret algorithms to form a Nash equilibrium in the metagame. In an effort to characterize which types of games playing no-swap-regret algorithms is "optimal", the authors show that if a game G does not contain a pure Nash equilibria, then there does not exist a pair of no-swap-regret algorithms which form a Nash equilibrium in the metagame, when player utility functions in G are randomly perturbed.

Towards answering (3), the authors show that there exists a no-(external)-regret algorithm for player B and a strategy for player A such that the average reward for player A converges to their best possible feasible reward. However, the authors show that against any mean-based no-regret learner (a popular subset of no-external-regret algorithms), there does not exist a strategy for player A which can get "close to" the best possible feasible reward.

Finally, the authors answer (4) by showing how to learn the Stackelberg strategy by simulating best-response queries against the no-regret learner. While they show that in general this may require exponentially-many queries, polynomially-many queries are sufficient if the no-regret learner is playing a no-adaptive-regret algorithm.

**Strengths:**

While the authors are not the first to consider the general problem of playing a repeated game against a no-regret learner, this paper both introduces and addresses a (very) wide range of important and well-motivated questions surrounding the topic. The results contained in this submission provide valuable insights into this highly nuanced problem. While no one result stands out in particular, the sheer breadth of the results obtained by the authors in this submission is very impressive and the submission as a whole presents the clearest picture to-date of "the right thing to do" when playing against a no-regret learner.

**Weaknesses:**

With that being said, the breadth of the results obtained by the authors makes it unclear what the main takeaway of the submission should be. At times, the submission reads like a laundry list of results about playing against a no-regret learner. Additionally, a longer discussion on related works in the main body (particularly (10) and (22)) would help someone who is not as familiar with the area better understand the main contributions of the authors.

**Questions:**

In Section 3, why is Nash equilibrium the "right" solution concept for the metagame?

**Limitations:**

The authors have adequately addressed the limitations of their work.

---

> ### Author Rebuttal · Authors · 2023-08-04
>
> Thank you for your feedback! Indeed, we view our results as indicating that the answer to the question of “what should you do when playing against a no-regret learner?” is very much “it depends”, and we aim to explore several branches of this decision tree (is their algorithm known? is the game known? etc.). A second-order takeaway is perhaps that the Stackelberg value is often a “reasonable” benchmark to target. We will certainly use the additional provided page in the camera-ready version to further clarify the relevant background and the narrative thread of our results.
>
> As for the meta-game, the question of alternate possible solution concepts is definitely an interesting one. We focused on Nash equilibria because the meta-game is essentially a “one-round game” where players act independently (by committing to an algorithm/strategy), and here Nash equilibria exactly capture whether players have an incentive to deviate up-front from committing to a specific learning algorithm.

---

> > ### Comment · Reviewer_vLcT · 2023-08-11
> >
> > Thanks for the reply. I have read the authors' rebuttal.

---

### Official Review · Reviewer_hPuK · 2023-07-06

**Soundness:** 3 good
**Presentation:** 3 good
**Contribution:** 3 good
**Rating:** 6
**Confidence:** 5

**Summary:**

The paper explores tradeoffs between reward and regret in repeated gameplay between two agents. It introduces a concept of generalized equilibrium that allows for different regret constraints, resulting in feasible values for each agent. The paper shows that such equilibria can be reached by algorithms maintaining regret guarantees against any opponent. The paper also examines tradeoffs in terms of the opponent's algorithm choice and characterizes the maximal reward achievable against a no-regret learner. It demonstrates that different classes of no-regret algorithms can lead to varying rewards.

**Strengths:**

- Theoretical analysis is solid and convincing.
- The problem studied is interesting. Although running no-regret dynamics leads to CCE, no work attempts to doubt "running no-regret" this thing itself.


**Weaknesses:**

I think some realistic running examples can be supplemented for better illustration.

**Questions:**

See in Weaknesses

**Limitations:**

See in Weaknesses

---

> ### Author Rebuttal · Authors · 2023-08-04
>
> Thanks for your comments! While our focus for the paper is intended to be primarily theoretical, in the appendix we give examples of games where we show explicit separations between feasible equilibrium values, and we are happy to explore simulating algorithms on these games.

---

### Official Review · Reviewer_TzAU · 2023-07-06

**Soundness:** 3 good
**Presentation:** 3 good
**Contribution:** 3 good
**Rating:** 7
**Confidence:** 2

**Summary:**

This paper addresses several interesting questions regarding the tradeoff between reward and regret in repeated gameplay between two agents. Three problems are sequentially investigated. 1. The paper provides a characterization of the setting when running a no-swap-regret learning algorithm is preferred over playing the Stackelberg strategy; it further showed that such a setting has measure zero and almost does not happen. 2. This paper shows that if the opponent is running any no-regret algorithm, the utility of the player is upper bounded by the unconstrained-external value of the game; such an upper bound is achievable for a particular no-regret algorithm of the opponent. 3. The paper shows that it is possible to convert any best-response query algorithm for finding Stackelberg equilibria via best-response queries to an adaptive strategy that learns Stackelberg equilibria via repeated play against a generic no-regret learner, albeit potentially at the cost of an exponential blow-up in the number of rounds.


**Strengths:**

The several questions addressed in this paper are very interesting. In algorithmic game theory, many existing work focus on algorithms for finding an equilibrium via repeated gameplay. However, in repeated gameplay, the agent’s interest is often maximizing reward and has the motivation to deviate from the regret minimization algorithm. This paper studies when deviating from the regret minimization algorithm is beneficial and when it is not.


**Weaknesses:**

There are still many unsolved open questions. For example, for specific no-regret algorithm classes, it is not clear how much one can exploit.


**Questions:**

NA

---

> ### Author Rebuttal · Authors · 2023-08-04
>
> Thanks for the review! Indeed there are several remaining open questions that we think would be interesting to explore in future work.

---

### Official Review · Reviewer_o15k · 2023-07-06

**Soundness:** 4 excellent
**Presentation:** 4 excellent
**Contribution:** 4 excellent
**Rating:** 8
**Confidence:** 5

**Summary:**

This paper considers equilibria between agents that have arbitrary regret benchmarks (corresponding to different equilibrium concepts), and the relation between those equilibria and the interactions of no-regret learning agents.  It is known in the literature that an agent who knows that the other is playing a no-external-regret algorithm can guarantee themself the Stackelberg leader value (i.e., $Val_A(\emptyset,\mathcal{E})$ in the terminology of this paper).  This paper extends that results by showing that in fact it can be better to play a no-external-regret learning strategy against a no-swap-regret learner (i.e., a learning algorithm with a weaker regret guarantee can get higher utility, all else being equal).  A further result is that any generalized equilibrium is reachable by a pair of regret-minimizing learning strategies.

The paper also considers the complexity of learning Stackelberg strategies; it turns out to be easier to learn a Stackelberg leader strategy against a learning algorithm with a stronger regret guarantee, because it reacts more quickly to best-respond to changes in the other player's behavior.




**Strengths:**

This is an extremely strong paper.  The question of when minimizing regret benchmarks also lead to good performance (the thing that we actually care about, in general) is really important, and this paper provides rigorous, compelling, and general answers.  The generalized equilibrium framework and the connections between learning and equilibrium are very clear and likely to have a significant impact on the learning in games literature.  I strongly expect to refer to this paper in the future.

The complexity results at the end are a nice touch as well.




**Weaknesses:**

I have no major complaints about the paper.  Here are some minor comments/issues:

- I was initially very surprised by Theorem 1; a little more hand-holding about why this doesn't contradict Barman & Ligett [2015] might have helped.  (Basically, the result doesn't require the algorithm pair to _find_ an optimal equilibrium, instead it will converge without regret to an exogenously _specified_ equilibrium).
- p.6: "Further, each equilibrium set can be optimized over via a linear program.": I'm not sure what this means.
- p.8: "Let $\sigma_{i,t}$ be the cumulative reward resulting from playing action $i$ for the first $t$ rounds": It would be clearer to avoid the notational collision with strategies by using a different letter
- p.8: A few more details in the proof sketch for Theorem 5 would be helpful; it took me a while to convince myself even that the statement made sense.


[Barman & Ligett 2015]: "Finding Any Nontrivial Coarse Correlated Equilibrium Is Hard"



**Questions:**

none

---

> ### Author Rebuttal · Authors · 2023-08-04
>
> Thank you for your comments!
>
> You are correct that Theorem 1 applies to exogenously specified equilibria, and does not require “learning” the target equilibrium on the fly. However, our focus is on two-player games, for which it is also possible to efficiently optimize over each of the equilibrium sets we consider via linear programming (as we show in Proposition 3). Even for multiplayer games, one can always compute an “optimal” (e.g. in welfare, or for any player) (C)CE via a linear program which is polynomial in the size of the game, with variables for the probability of each action profile and linear constraints for each player-action(-action) pair which enforce regret constraints. The size of this LP is polynomial in {# actions} but exponential in {# players}, as the normal-form representation of a many-player game has exponentially many {# players}. In contrast, the results of [BK15] apply to “succinct” games with structured rewards (e.g. routing games, see [PR08] for many other examples) in which rewards are fully determined by a representation which is polynomial in {# players, # actions}, and as such are not in contradiction with the LP approach which uses a normal-form representation.
>
> We will clarify this in the paper, and will also fix the notation overload (thanks for catching this!) as well as further clarify the intuition behind Theorem 5; the key idea there is showing that mean-based algorithms are strictly stronger than “worst-case” no-regret algorithms, and thus are harder to exploit.
>
> [PR08] Computing correlated equilibria in multi-player games - Papadimitriou & Roughgarden

---

> > ### Comment · Reviewer_o15k · 2023-08-14
> >
> > Thanks for the additional clarifications!

---

### Decision · Program_Chairs · 2023-09-21

**Decision:**

Accept (spotlight)

**Comment:**

This paper lools at the interplay of no-regret learning and convergence in games.

All the reviewers, and myself, enjoyed it  and is therefore strongly recommend for acceptance.